# Joint Training of Deep Ensembles Fails Due to Learner Collusion

**Alan Jeffares**
University of Cambridge
aj659@cam.ac.uk

**Tennison Liu**
University of Cambridge
tl522@cam.ac.uk

**Jonathan Crabbé**
University of Cambridge
jc2133@cam.ac.uk

**Mihaela van der Schaar**
University of Cambridge
mv472@cam.ac.uk

## Abstract

Ensembles of machine learning models have been well established as a powerful method of improving performance over a single model. Traditionally, ensembling algorithms train their base learners independently or sequentially with the goal of optimizing their joint performance. In the case of deep ensembles of neural networks, we are provided with the opportunity to directly optimize the true objective: the joint performance of the ensemble as a whole. Surprisingly, however, directly minimizing the loss of the ensemble appears to rarely be applied in practice. Instead, most previous research trains individual models independently with ensembling performed *post hoc*. In this work, we show that this is for good reason *- joint optimization of ensemble loss results in degenerate behavior*. We approach this problem by decomposing the ensemble objective into the strength of the base learners and the diversity between them. We discover that joint optimization results in a phenomenon in which base learners collude to artificially inflate their apparent diversity. This pseudo-diversity fails to generalize beyond the training data, causing a larger generalization gap. We proceed to comprehensively demonstrate the practical implications of this effect on a range of standard machine learning tasks and architectures by smoothly interpolating between independent training and joint optimization.[1]

## 1 Introduction

Deep ensembles [1] have proven to be a remarkably effective method for improving performance over a single deep learning model. This success has been attributed to deep ensembles exploring multiple modes in the loss landscape [2], in contrast to variational Bayesian methods, which generally attempt to better explore a single mode [e.g. 3]. Recent work has proposed an alternative view suggesting that deep ensembles' success can be explained as being an effective method of scaling smaller models to match the performance of a single larger model [4]. Regardless of the specific mechanism, deep ensembles' empirical success has been at the heart of numerous state-of-the-art deep learning solutions in recent years [e.g. 5, 6, 7].

Historically, non-neural network ensembles such as a bagging predictor [8] or a random forest [9] were generally trained independently and aggregated at test time. Sequential training methods such as boosting [10] were also developed to produce more collaborative ensembles. Training individually and evaluating jointly was sensible for these methods, in which the base learners were typically

---

[1]Code is provided at https://github.com/alanjeffares/joint-ensembles.

decision trees that *require* individual training. Formally, for a single example $\mathbf{x}$, a loss function $\mathcal{L}$ and denoting the prediction of learner $j$ on $\mathbf{x}$ as $\mathbf{f}_j \in \mathbb{R}^d$ with target $\mathbf{y} \in \mathbb{R}^d$, we define the *joint objective* as $\mathcal{L}(\frac{1}{M}\sum_{j=1}^{M}\mathbf{f}_j, \mathbf{y})$ and the *independent objective* as $\frac{1}{M}\sum_{j=1}^{M}\mathcal{L}(\mathbf{f}_j, \mathbf{y})$.[2] Interestingly, modern deep ensembles are also generally trained independently and evaluated jointly. This is less intuitive as the performance of the ensemble as a whole (i.e. the joint objective) is the true objective of interest and jointly training a deep ensemble is quite natural for the backpropagation algorithm. Indeed, one might expect there to be no need for the proxy objective of independent training in this case, as the joint objective *should* find a solution in which individual learners collaborate to minimize the ensemble's loss. In practice, however, independent training generalizes better than joint training in this context. A phenomenon that we demonstrate in Section 3 and investigate throughout this work.

In Section 4, we demonstrate that ensemble diversity is the foundation upon which understanding the limitations of joint training sits. We show that any twice differentiable joint loss function of an ensemble can be decomposed into the individual losses and a second term, which can be interpreted as ensemble diversity. This diversity term encourages some level of ambiguity among the base learners' predictions, a generalization of well-established work in the regression setting [11]. Mathematically, this term is exactly the difference between independent and joint training and, therefore, must encapsulate the disparity in performance. In Section 5, we introduce a scalar weighting on the diversity term allowing us to smoothly interpolate between the extremes of fully independent and fully joint training.

Then, in Section 6, we diagnose and empirically verify a precise cause of the sub-optimal performance of jointly trained ensembles. We show that the additional requirement for diversity in the training objective can be trivially achieved by an adverse effect we term *learner collusion*. This consists of a set of base learners artificially biasing their outputs in order to trivially achieve diversity while ensuring that these biases cancel in aggregate as illustrated in Figure 1. We perform extensive experiments verifying this hypothesis. We then show that although training loss can be driven down by learner collusion, this does not generalize beyond the training set, resulting in a larger generalization gap than independent training. Finally, we examine the practical implications of learner collusion on standard machine learning benchmarks.

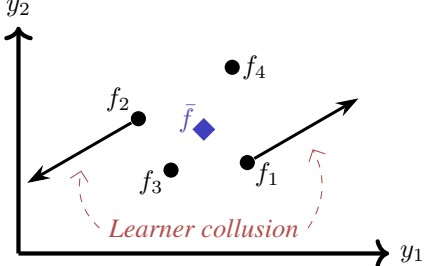

Figure 1: **Learner collusion.** An illustrative example in which a subset of base learners ($f_1$ & $f_2$) *collude* in output space by shifting their predictions in opposing directions such that the ensemble prediction ($\bar{f}$) is unchanged, but diversity is inflated.

This work addresses the pervasive yet relatively understudied phenomenon that *directly optimizing a deep ensemble fails*. **(1)** We unify various strands of the ensemble literature to provide a deep, practical categorization of ensemble training and diversification. **(2)** We present and verify a comprehensive explanation for this counter-intuitive limitation of deep ensembles with both theoretical and empirical contributions. **(3)** We investigate the catastrophic consequences of this issue on practical machine learning tasks. As this investigation takes a somewhat non-standard format, we provide an abridged summary of this work and its technical contributions in Appendix A.

## 2   Background

**Independent training.** This refers to ensembles[3] in which *each base learner is optimized independently, with no influence from the other ensemble members either explicit or implicit*. This includes many non-neural ensemble techniques in which diversity between the base learners results in ensemble performance better than any individual. This diversity is often achieved by sampling the input space [e.g. bagging in 8] and the feature space [e.g. random forest in 9]. In the case of deep ensembles of neural networks, independent training is also generally applied, with diversity achieved using random weight initialization and stochastic batching [1, 2]. Interestingly, bootstrap sampling of the inputs is ineffective in the case of deep ensembles [12]. Several notable state-of-the-art results

---

[2]In Section 4 we further generalize this definition to account for (a) non-equal weighting between the predictions of ensemble members and (b) aggregating ensemble outputs before normalization (i.e. *score averaging*).

[3]In this work we consider ensembles in the least prescriptive sense (i.e. any algorithm that aggregates multiple target predictions).

have been achieved by ensembling several independently trained but individually performant neural networks. For example, the groundbreaking results in the ImageNet challenge were often achieved by independently trained ensembles such as VGG [13] GoogLeNet [5], and AlexNet [14]. Seq2Seq ensembles of long short-term memory networks [15] achieved remarkable performance for neural translation. More recently in the tabular domain, ensembles of transformers have outperformed state-of-the-art non-neural methods [7].

**Ensemble-aware individual training.** This refers to the case where *each base learner is still being optimized as an individual, but with some additional technique being applied to coordinate the ensemble members*. A prominent approach is sequentially trained, boosting ensembles in which each new member attempts to improve upon the errors of the existing members (e.g. AdaBoost [10] & gradient boosting [16]). In the case of deep ensembles, an alternative approach consists of training each base learner with a standard individual loss with the addition of an ensemble-level regularization term designed to manage diversity. This often consists of an f-divergence between the learners' predictive distributions. Examples include the pairwise $\chi^2$-divergence [17], $KL$-divergence, but between the learner and the ensemble [18], and a computationally efficient, kernelized approximation of Renyi-entropy [19]. Another prominent regularization approach is based on the seminal idea of negative correlation learning (NCL) which penalizes individual learners through a correlation penalty on error distributions [20, 21]. This penalty weakens the relationship with other learners and controls the trade-off between bias, variance, and covariance in ensemble learning [22]. Recent works have adapted NCL for improved computational efficiency with large base learners [23] and attempt to generalize NCL beyond the regression setting [24]. [25] encourages ensemble diversity in weight and functional space using a "repulsive" term while other works show that a single learner can be viewed as an ensemble with diversity encouraged implicitly (e.g. [26, 27]). The efficient BatchEnsemble [28] can also be characterized as an ensemble-aware individual training approach.

**Joint training.** This refers to the case where *the ensemble's predictions are optimized directly as if it were a single model*. Intuitively, one might naturally expect joint training to outperform the previously discussed methods as it directly optimizes (at training time) the test metric of all three approaches - *ensemble loss*. Empirically, however, this is not the case as joint training achieves inferior performance in general. [29] reported poor results when joint training averages the base learners' output scores but excellent performance when they are averaged at the log-probability or loss levels. However, later work showed that the two alleged success cases were, in fact, implementing independent training rather than joint training unbeknownst to the authors [30]. This later work also reported poor generalization for score-averaged ensembles. We provide a detailed clarification and gradient analysis in Appendix E. Elsewhere, [31] experienced poor performance from joint training for both score and probability averaging. Concurrently to this work, [32, 33] also found that joint training consistently resulted in worse empirical performance. To put the matter to rest, we demonstrate these issues with joint training in the next section.

## 3 Empirically Evaluating Joint Training

In this section, we briefly demonstrate the effect that we investigate throughout this work. That is, when evaluating the ensemble loss at test time, optimizing for that same ensemble loss objective at training time (i.e. joint training) is *less* effective than optimizing each of the base learners independently (i.e. independent training). We do this by performing a straightforward comparison of the two methods. We compare the test set performance of independent and joint training on ImageNet [34]. We consider several architectures as base learners of an ensemble of various sizes determined by computational limitations. We report test set top-1 and top-5 accuracy for both independently and jointly trained ensembles. These results are included in Table 1. Complete experimental details for all experiments in this work are provided in Appendix B.

Here, we observe the counter-intuitive phenomenon that independently trained ensembles consistently outperform jointly trained ensembles, despite all models being evaluated using the joint objective. This is consistent with previous works that have attempted to apply joint training [30, 31]. In the following sections, we will proceed to investigate and explain this effect through the lens of ensemble diversity.

**Summary:** Joint training performs considerably worse than independent training in practice.

Table 1: **Joint training on ImageNet.** We compare independent to joint training across a range of architectures trained from scratch on the ImageNet dataset. Top 1 & 5 test accuracy is reported for both methods, where independent training consistently performs better.

| Base learner | Number of learners | Independent | | Joint | |
| --- | --- | --- | --- | --- | --- |
| | | Top-1 | Top-5 | Top-1 | Top-5 |
| ResNet-18 [35] | 5 | **68.10**% | **88.43**% | 62.74% | 84.03% |
| ResNet-34 [35] | 3 | **73.11**% | **91.25**% | 68.91% | 88.01% |
| DenseNet [36] | 3 | **70.62**% | **89.51**% | 60.81% | 81.35% |
| ShuffleNet V2 (0.5×) [37] | 10 | **61.00**% | **81.94**% | 53.41% | 74.88% |
| ShuffleNet V2 (1×) [37] | 5 | **68.56**% | **87.64**% | 61.96% | 81.56% |
| SqueezeNet V1.1 [38] | 5 | **56.25**% | **78.65**% | 42.69% | 65.24% |
| MobileNet V3-small [39] | 5 | **67.43**% | **86.82**% | 57.34% | 78.30% |
| MobileViT [40] | 3 | **61.29**% | **82.84**% | 54.69% | 76.79% |

# 4 Diversity in Ensemble Learning

To understand the limitations of joint training, we will need to examine the joint objective through the lens of diversity. In this section, we argue for a unified definition of diversity which generalizes much of the existing literature on the subject.

## 4.1 Preliminaries

We begin with some notation. We consider ensembles made up of $M$ individual learners. The $j$'th individual learner maps an input $\mathbf{x} \in \mathbb{R}^p$ to a $d$-dimensional output space such that $\mathbf{f}_j : \mathbb{R}^p \to \mathbb{R}^d$ and attempts to learn some regression target(s) $\mathbf{y} \in \mathbb{R}^d$ or, in the case of classification, labels which lie on a probability simplex $\Delta^d$. Without loss of generality, we can refer to $\mathbf{y} \in \mathcal{Y}$. We primarily consider combining individual learner predictions into a single ensemble prediction as a weighted average $\sum_{j=1}^{M} w_j \mathbf{f}_j(\mathbf{x}) = \bar{\mathbf{f}}(\mathbf{x})$ where $w_j \in \mathbb{R}$ denotes the scalar weights. Although not strictly required for much of this work, we will generally assume that these weights satisfy the properties $\sum_{j=1}^{M} w_j = 1$ and $w_j \geq 0 \ \forall \ j$, ensuring that each individual learner can be interpreted as predicting the target. Scalars are denoted in plain font and vectors in bold. We generally drop the dependence on $\mathbf{x}$ in our notation such that individual and ensemble predictions are denoted $\mathbf{f}_j$ and $\bar{\mathbf{f}}$ respectively. Finally, we formalize the respective losses of the ensemble and the base learners in Definition 4.1.

**Definition 4.1** (Ensemble and Individual Loss). For a given loss function $\mathcal{L} : \mathcal{Y}^2 \to \mathbb{R}^+$, we define:

(a) The overall ensemble loss $\mathrm{ERR} \coloneqq \mathcal{L}(\bar{\mathbf{f}}, \mathbf{y})$.

(b) The weighted individual loss of the base learners $\overline{\mathrm{ERR}} \coloneqq \sum_{j=1}^{M} w_j \mathcal{L}(\mathbf{f}_j, \mathbf{y})$.

Intuitively, ERR measures the predictive performance of the ensemble, while $\overline{\mathrm{ERR}}$ measures the aggregate predictive performance of the individual ensemble members.

## 4.2 A Generalized Diversity Formulation

**Diversity in regression ensembles.** The prevailing view of diversity in regression ensembles began in [11], which showed that the mean squared error of a regression ensemble can be decomposed into the mean squared error of the individual learners and an additional *ambiguity* term which measures the variance of the individual learners around the ensemble prediction showing

$$\sum_j w_j (f_j - \bar{f})^2 = \sum_j w_j (f_j - y)^2 - (\bar{f} - y)^2,$$

$$\mathrm{DIV} = \overline{\mathrm{ERR}} - \mathrm{ERR}. \tag{1}$$

This decomposition provides an intuitive definition of diversity whilst also showing that a jointly trained ensemble, that is one that simply optimizes ERR, encourages diversity by default due to it being *baked into the objective*. Later work from the evolutionary computation literature proposed an apparently alternative measure of diversity called Negative Correlation Learning, which argues

that lower correlation in individual errors is equivalent to higher diversity [21]. The apparent disagreement between these two seemingly valid views on diversity was resolved when it was shown that they are in fact equivalent up to a scaling factor of the diversity term [22].

**Generalized diversity.** Motivated by the regression case, we will argue that the difference between the individual and ensemble errors has a natural interpretation as the diversity among the predictions of the ensemble members *for any twice (or more) differentiable loss function*. In fact, this generalizes several existing notions of diversity proposed throughout the literature for specific cases. Specifically, we define diversity as in Definition 4.2 and will proceed to justify this definition in what follows.

**Definition 4.2** (Diversity). The diversity of an ensemble of learners (DIV) is the difference between the weighted average loss of individual learners and the ensemble loss:

$$\mathrm{DIV} := \overline{\mathrm{ERR}} - \mathrm{ERR}$$

**Diversity in classification ensembles.** It is natural to next verify that this definition is appropriate in the classification setting. We proceed by concretely deriving the diversity term in this setting. There are two standard methods for aggregating base learners' predictions into a single ensemble prediction in classification. Averaging at the score (preactivation) level and averaging at the probability level. We address each of these in turn.

**①** *Score averaging.* We consider a classification task with target distribution $\mathbf{y} \in \mathcal{Y} \in \Delta^d$ with $d$ classes, and consider ensembling by averaging the base learners' scores, before applying the normalizing map $\phi$ (typically a softmax). We define $\mathbf{p}_j : \mathcal{X} \to \Delta^d \equiv \phi \circ \mathbf{f}_j$ as the normalized predictive distribution for learner $j$ and $\bar{\mathbf{q}} = \phi \circ \bar{\mathbf{f}}$ the predictive distribution for the ensemble, where $\bar{\mathbf{f}} = \sum_{j=1}^M w_j \mathbf{f}_j$ as before. In this setting, it can be shown [41] that

$$\mathrm{DIV} = \sum_{j=1}^M w_j D_{KL}(\mathbf{y}||\mathbf{p}_j) - D_{KL}(\mathbf{y}||\bar{\mathbf{q}}) = \sum_{j=1}^M w_j D_{KL}(\bar{\mathbf{q}}||\mathbf{p}_j),$$

where $D_{KL}$ is the $KL$-divergence. Intuitively, this diversity term measures the weighted divergence between each ensemble member's predictive distribution and the full ensemble's predictive distribution, a sensible expression for diversity. It should be noted that this decomposition holds for normalizing functions beyond the softmax [30]. Specifically, this exact expression holds for all exponential family normalization functions with score averaging. This allows the diversity decomposition described in this work to apply to disparate settings as demonstrated by its recent application in modeling diversity among the Poisson spike train outputs of spiking neural networks [42].

**②** *Probability averaging.* This setting diverges from score averaging by setting the ensemble predictive distribution $\bar{\mathbf{p}}$ to be the average of the base learners' *probabilities* such that $\bar{\mathbf{p}} = \sum_{j=1}^M w_j(\phi \circ \mathbf{f}_j)$. This is a more natural strategy, averaging the scale-invariant probabilities while also providing each base learner with unique gradients during backpropagation (see a detailed gradient analysis in Appendix E). The resulting expression for diversity is included in Theorem 4.3.

**Theorem 4.3.** *For a probability averaged ensemble classifier, trained using cross-entropy loss $\mathcal{L}_{CE}$ with $k^\star \in \mathcal{K}$ denoting the ground truth class, diversity is given by*

$$\mathrm{DIV} = \sum_{j=1}^M w_j \mathcal{L}_{CE}(\mathbf{y}, \mathbf{p}_j) - \mathcal{L}_{CE}(\mathbf{y}, \bar{\mathbf{p}}) = \sum_{j=1}^M w_j \cdot y(k^\star) \log \frac{\bar{p}(k^\star)}{p_j(k^\star)}$$

*Proof.* Please refer to Appendix C.3. □

*Remark* 4.4. It is not immediately obvious that this expression has a natural interpretation as diversity. However, in Appendix C.3 we show it to be well aligned with the notion of diversity used throughout this work by analysis of the distribution of $p_j(k^\star)$ around $\bar{p}(k^\star)$.

Beyond the KL-divergence family, alternative loss functions also result in sensible diversity expressions. Brier score, expressed as $(\bar{\mathbf{p}} - \mathbf{y})^T(\bar{\mathbf{p}} - \mathbf{y})$, can be shown to obtain $\mathrm{DIV} = \sum_j w_j(\bar{\mathbf{p}}_j - \mathbf{p})^T(\bar{\mathbf{p}} - \mathbf{p}_j)$, the variance of the base learner probabilities around the ensemble probabilities [4].

For 0-1 loss and the case of majority voting, diversity is also measured by the disagreement between the (categorical) predictions of the base learners and the ensemble prediction [43].

**Diversity in a general setting.** The previous paragraphs verified the diversity definition introduced in Definition 4.2 to be a sensible interpretation for several common loss functions. A more general question can be asked — *Is* DIV *reasonable for any choice of loss functions?* The answer is *yes*, and this turns out to be no coincidence. To demonstrate why, we analyze the properties of the diversity expression for *any* twice differentiable loss function and discover a general form that permits a clear interpretation as diversity. To do so, we present Theorem 4.5, a multidimensional generalization of one proposed in [44] which, in turn, generalizes our notion of diversity and analysis of joint training to practically any loss function.

**Theorem 4.5.** *[Diversity for General Loss] For any loss function* $\mathcal{L} : \mathcal{Y}^2 \to \mathbb{R}^+$ *with* $\dim(\mathcal{Y}) \in \mathbb{N}^*$ *that is at least twice differentiable, the loss of the ensemble can be decomposed into*

$$\mathrm{DIV} = \frac{1}{2} \sum_{j=1}^{M} w_j \, (\mathbf{f}_j - \bar{\mathbf{f}})^\intercal \cdot \mathcal{H}_f^{\mathcal{L}}(\mathbf{f}_j^*, \mathbf{y}) \cdot (\mathbf{f}_j - \bar{\mathbf{f}})$$

$$\text{with } \mathbf{f}_j^* = \bar{\mathbf{f}} + c_j(\mathbf{f}_j - \bar{\mathbf{f}}),$$

*where* $\mathcal{H}_f^{\mathcal{L}}$ *is the Hessian of* $\mathcal{L}$ *with respect to its first argument and some* $c_j \in [0, 1]$ *for all* $j \in [M]$.

*Proof.* Please refer to Appendix C.2. $\square$

Here, the measure of diversity is obtained by performing a Taylor expansion on the loss of the output of $\mathbf{f}_i$ near the ensemble output $\bar{\mathbf{f}}$ and using Lagrange's remainder theorem to obtain an *exact equality*. Additionally, $\mathbf{f}_i^*$ is an uncertain number, which approaches a constant in the limit. This diversity admits the same interpretation as the variance of learners around the ensemble, but is now weighted by a term dependent on the second derivative of the loss function.

One can easily verify that this generalized definition encompasses the uni-dimensional MSE ambiguity decomposition in Equation (1), where $\mathcal{L}''(f_i^*, y) = 2$. Similarly, the multidimensional MSE $\mathcal{L}(\mathbf{f}, \mathbf{y}) = \|\mathbf{f} - \mathbf{y}\|_2^2$ leads to the Hessian $\mathcal{H}_f^{\mathcal{L}} = 2\mathbf{I}$, where $\mathbf{I}$ is the identity matrix on $\mathcal{Y}$. From this, we get the multi-dimensional generalization $\mathrm{DIV} = \sum_j w_j \|\mathbf{f}_j - \bar{\mathbf{f}}\|^2 \geq 0$. Importantly, diversity has the same non-negative property ($\mathrm{DIV} \geq 0$) for any loss function with semi-positive definite Hessian $\mathcal{H}_f^{\mathcal{L}} \succeq 0$, such that ensemble performance is always better than individual learner performance. Another typical example is the classification loss $\mathcal{L}(\mathbf{f}, \mathbf{y}) = -\mathbf{y}^\intercal \log \mathbf{f}$. Here, it is trivial to show that $\mathcal{H}_f^{\mathcal{L}}(\mathbf{f}, \mathbf{y}) = \mathrm{diag}(\mathbf{y} \oslash \mathbf{f}^2)$, where $\oslash$ denotes the component-wise division. Again, this Hessian matrix is positive semi-definite for $\mathbf{y}, \mathbf{f} \in \Delta^d$ so that $\mathrm{DIV} \geq 0$. Revisiting our previous question, we verify that, for *at least twice-differentiable* loss functions (e.g. logistic loss: $\mathcal{L} = \log(1 + \exp^{-yf})$ & exponential loss: $\mathcal{L} = \exp^{-yf}$), DIV admits a generalized interpretation as the variance of individual predictors around ensemble prediction. This makes a robust case for Definition 4.2, which provides a principled estimate of diversity (as opposed to heuristic measures) and can be easily obtained by calculating the difference between ensemble loss and weighted average learner loss.

**Summary:** The difference between independent loss and joint loss has a natural interpretation as diversity, providing a robust generalization of much of the existing literature.

## 5  An Augmented Objective

Given this generalized definition of diversity, one might naturally consider placing a scalar weighting on its contribution, resulting in an augmented training objective. In fact, such an approach has already been applied for special cases of diversity including regression [22], score averaging [30], and, concurrently to this work, in [33, 32]. Throughout our experiments, we will consider this augmented objective in Equation (2), allowing us to analyze the effects of smoothly transitioning between independent and joint training.

$$\mathcal{L}^\beta(\bar{\mathbf{f}}, \mathbf{y}) \coloneqq \overline{\mathrm{ERR}} - \beta \cdot \mathrm{DIV}$$
$$= (1 - \beta) \cdot \overline{\mathrm{ERR}} + \beta \cdot \mathrm{ERR}. \tag{2}$$

When $\beta = 0$, this amounts to training each learner independently (i.e. *independent training*) and $\beta = 1$ corresponds to optimizing the loss of the ensemble as a whole (i.e. *joint training*). Clearly, intermediate values of $\beta$ can be interpreted as interpolating between the two.

When examining Equation (2), one might be tempted to encourage more diversity by setting $\beta > 1$. Previous work by Brown et al. [45] showed that, in the case of regression, optimization can become degenerate for larger values of $\beta$, proving an upper bound of $\beta < \frac{M}{M-1}$. More recently, [30] showed that in the case of score averaged ensembles using their *modular loss*, the ensemble loss ERR was unbounded from below for $\beta > 1$ – i.e. DIV can trivially be increased faster than its corresponding increase in $\overline{\text{ERR}}$, thus (superficially) minimizing the loss. In Theorem 5.1 we generalize this same bound on $\beta$ to *all* loss functions and further include the case of probability-averaged ensembles.

**Theorem 5.1.** *[Pathological Behavior for $\beta > 1$] We consider a target set $\mathcal{Y} = \mathbb{R}^d$ or $\mathcal{Y} \subset \mathbb{R}^d$ being a compact and convex subset of $\mathbb{R}^d$. Let $\mathcal{L} : \mathcal{Y}^2 \to \mathbb{R}^+$ be a continuous and finite function on the interior $\text{int}(\mathcal{Y}^2)$. We assume that the loss is not bounded from above on $\mathcal{Y}^2$ in such a way that there exists $\mathbf{y} \in \text{int}(\mathcal{Y})$ and a sequence $(\mathbf{y}_t)_{t\in\mathbb{N}} \subset \mathcal{Y}$ such that $\mathcal{L}(\mathbf{y}_t, \mathbf{y}) \to +\infty$ for $t \to \infty$. If $\beta > 1$, then the augmented loss $\mathcal{L}^\beta$ defined in Equation (2) is not bounded from below on $\mathcal{Y}^2$.*

*Proof.* Please refer to Appendix C.4. □

*Remark* 5.2. Note that we consider the case where $\mathcal{Y}$ is a compact and convex subset of $\mathbb{R}^d$ to include classifiers, for which the target set is a probability simplex $\Delta^{d-1}$.

As a consequence of the above theorem, one needs to impose the tighter bound $\beta \leq 1$ in order to have a well-defined optimization objective. Furthermore, intermediate values where $\beta \in (0, 1)$ may have an additional interpretation beyond interpolating between independent and joint training. In Appendix D we show that, in the regression setting, $\mathcal{L}^\beta$ is exactly equivalent to applying independent training while adjusting the targets to account for the errors of the ensemble as measured by its gradient. We derive an exact equivalence between $\beta$ in $\mathcal{L}^\beta$ and the magnitude of the adjustment of the target in this gradient-adjusted target setting.

**Summary:** An interpolation between independent training and joint training is a well-motivated and useful objective but, to avoid pathological behavior, one should not extrapolate to higher diversity.

## 6  Why Does Joint Training Fail?

Given the universal interpretation of diversity developed in the previous sections paired with its exclusive impact on the joint training objective, we now turn our attention to answering the question - *why does joint training of deep ensembles perform worse than independent training?* Some previous works have attempted to partially address this question. [29] suggested that score averaging fails as the softmax function "distorts" the averages. However, little evidence exists to support this claim. When faced with the same issue, [31] suggested the issue could be due to score averaged ensemble members receiving identical gradient values. We address this point in a gradient analysis in Appendix E, where we (a) point out that the optimal solution to joint objective would still achieve equal or lower loss for score averaging, and (b) show that probability averaged ensemble members receive variable gradient expressions and *still* suffer from poor performance. Finally, [30] noticed vastly different test set performances among the base learners which they referred to as *model dominance*. Indeed, this is a symptom of the underlying issue with joint training which we describe and verify throughout the rest of this section. The explanation we propose for the poor empirical results achieved by joint training consists of two claims:

*Claim 1:* the additional diversity term in the joint learning objective (Equation (2)) results in a dual objective. The DIV term can always be artificially inflated if multiple learners collude to bias their predictive distributions in opposing directions which cancel out in aggregate and, therefore, is easily maximized across the training data. We refer to this phenomenon as *learner collusion*.

*Claim 2:* while learner collusion can still minimize the loss on the training data, this solution may contain little genuine diversity resulting in a solution that achieves worse generalization than independent training. In Appendix F, we provide a detailed illustrative example of learner collusion for the regression case from an optimization perspective.

In Section 6.1 we empirically demonstrate this learner collusion effect is emergent. Then, in Section 6.2, we show that the low training loss achieved by learner collusion generalizes poorly onto unseen data. Finally, in Section 6.3, we establish the practical implications of learner collusion on standard tasks.

### 6.1 Diagnosing Learner Collusion

In this section, we focus on claim 1 and empirically investigate the extent to which *learner collusion* does occur. Recall that the effect being investigated is that a subset of non-diverse base learners can trivially adjust their similar, or even identical, predictions to appear diverse (i.e. Figure 1). If learner collusion does indeed take place we suggest that the following three effects should be observable empirically: *(1)* the relative magnitude of diversity should be excessive relative to the ensemble loss, *(2)* if we remove constant output biases from the individual learners, the ensemble diversity should decrease significantly, and *(3)* learners should become highly dependent on each other such that removing a small subset of base learners is detrimental to the ensemble performance. We investigate each of these effects in turn.

We perform these experiments on four tabular datasets of the style upon which ensemble methods typically excel [46]. Here we focus on regression as it offers the most controlled setting to isolate the learner collusion phenomenon (see Appendix F), but in later experiments we demonstrate its effects in the classification setting too. We smoothly interpolate between independent and joint training using the augmented objective controlled by $\beta$ from Section 5. We report mean $\pm$ one standard deviation evaluated on a test set throughout. An extended description of the experimental protocol for these experiments is provided in Appendix B.

**Diversity explosion.** We begin with the most simple expected effect of learner collusion, that the relative magnitude of diversity is excessive with respect to the ensemble loss. The results are included in Figure 2 where we note a sharp increase in diversity for the joint objective. Specifically, we note that mean test set diversity remains relatively low for most values of $\beta$ but quickly explodes as $\beta \to 1$ to levels that are often significantly greater than the mean squared error of the ensemble predictions. This indicates that the base learners' variation around their own mean is much greater than the ensemble's variation around the label.

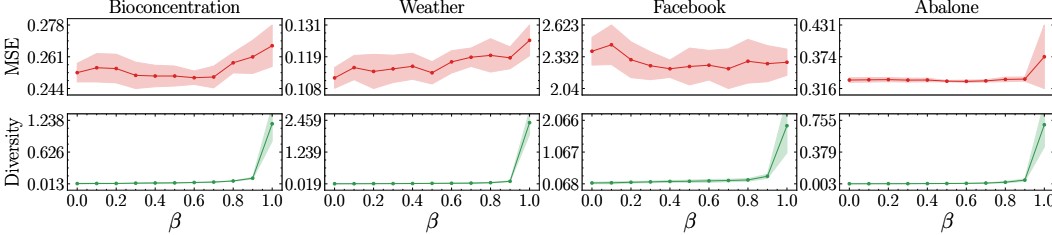

Figure 2: **Diversity explosion.** Test set diversity explodes relative to MSE across four datasets. We note two further empirical phenomena: (1) the non-linear relationship between $\beta$ and diversity which we discuss further in Appendix D and (2) the minimal effect diversity has on the test MSE on the Facebook dataset which is discussed in Appendix F.

**Debiased diversity.** We now proceed to the second postulated effect of learner collusion. Although learners could perform learner collusion on a per-example basis, the most straightforward form of this behavior would be learners that bias their predictions without any dependence on the example at hand. To investigate the extent to which this trivial form of collusion does occur we apply a post hoc debiasing to each of the individual learners in the ensemble which affects the trade-off between $\overline{\text{ERR}}$ and DIV without affecting ERR. Specifically, we adjust learner $j$'s predictions such that

$$f_j^{\text{debiased}}(\mathbf{x}_i) = f_j(\mathbf{x}_i) - b_j$$

$$= f_j(\mathbf{x}_i) - \left( \frac{1}{N} \sum_{i=1}^N f_j(\mathbf{x}_i) - \frac{1}{N} \sum_{i=1}^N \bar{f}(\mathbf{x}_i) \right).$$

In Appendix C.1 we prove that the debiasing term $b_j$ is the optimal diversity minimizer that keeps the ensemble predictions unchanged. In Figure 3 we show that applying this technique significantly reduces the ensemble's apparent diversity to far more reasonable values. In general, we find that this typically removes around half of the ensemble's diversity at the joint training region ($\beta = 1$). While example-dependent collusion could also emerge, these results already indicate that a significant proportion of the apparent diversity in jointly trained ensembles is due to an artificial biasing of outputs.

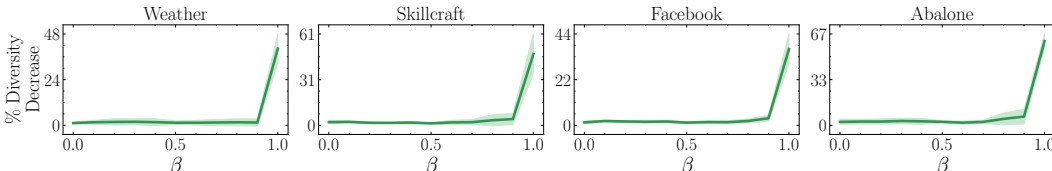

Figure 3: **Bias removal.** The percentage of diversity explained by the most simple form of learner collusion across each dataset. In the region where learner collusion appears to occur ($\beta \to 1$), a large proportion of the diversity can be explained by simple additive bias in the predictions.

**Learner codependence:** The final effect we predicted to emerge due to learner collusion is an excessive codependence among the base learners. If individual learners are engaging in learner collusion, the removal of a subset of these learners at test time should be catastrophic to the performance of the ensemble. This is because a learner producing excessive errors needs to be counterbalanced by other learners performing similar errors in the opposite direction, otherwise the ensemble predictions will be influenced by the poorly performing base learners. To test this we consider dropping a fraction of the individual learners at test time and evaluate the resulting ensemble performance. This is similar to experiments performed in [30], although from a very different perspective. In Figure 4 we present the results of randomly applying various levels of test time base learner dropout 100 times for each model and report the average resulting increase in test set error. These results clearly show that such a procedure is detrimental to ensemble performance at precisely the levels of $\beta$ at which we have hypothesized learner collusion to occur, while having only minimal impact elsewhere.

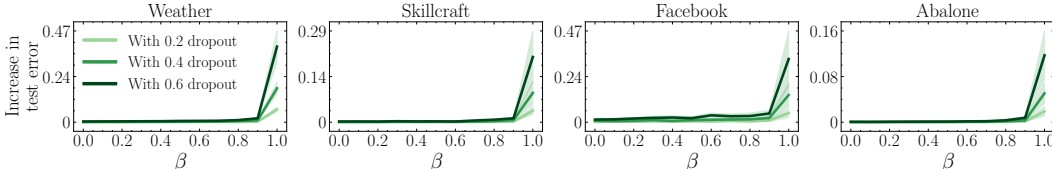

Figure 4: **Learner codependence.** Applying various levels of base learner dropout at test time and observing the relative increase in MSE. As $\beta \to 1$, dropping a subset of learners causes a greater increase in test error suggesting the occurrence of learner collusion.

**Summary:** Extensive empirical evidence confirms that training with the joint objective results in learner collusion.

## 6.2 Learner Collusion Generalizes Poorly

Given that we have shown that learner collusion emerges from joint training, we now proceed to examine the second claim. That is, although learner collusion does not prevent the training loss from being minimized, the solution it obtains will generalize more poorly than an independently trained ensemble with non-trivial diversity. This claim is more straightforward to investigate and only requires a comparison of the difference between the training and testing loss curves over the course of training. The difference between the two, which we refer to as the *generalization gap*, captures the drop in performance when a model transitions from training to testing data.

For this experiment, we return to the more standard machine learning benchmark of classification on the CIFAR tasks as introduced in Section 3. On each dataset we train a joint model (ERR) and an independent model ($\overline{\text{ERR}}$) and report the generalization gap of ensemble loss for both methods over the course of training. We repeat this six times and include a shaded region representing $\pm 2$ standard errors. The results are included in Figure 5 where we consistently find that joint training displays a significantly larger generalization gap as training proceeds. This is strongly suggestive of a much greater tendency towards overfitting in the jointly trained model while the independently trained model's generalization gap plateaus at a much lower level. We provide further analysis of the distinct training and testing curves in Appendix G.

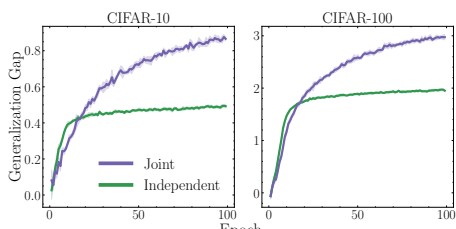

Figure 5: **Generalization gaps**. Joint training results in a larger generalization gap ($\mathcal{L}^{\text{test}}(\bar{\mathbf{f}}, \mathbf{y}) - \mathcal{L}^{\text{train}}(\bar{\mathbf{f}}, \mathbf{y})$) than independent training, despite both methods reporting test performance using the joint objective.

**Summary:** Learner collusion caused by joint training (ERR) results in greater overfitting exhibited by a larger generalization gap.

### 6.3 Practical Implications

We have now described the phenomenon of learner collusion caused by the joint training of deep ensembles, empirically verified its existence, and demonstrated how it results in poorer generalization. We complete this analysis with an empirical investigation of how learner collusion manifests in the ensemble performance on standard machine learning tasks. We use the augmented objective, $\mathcal{L}^{\beta}$ as described in Section 5, to examine the effects of smoothly increasing the level of joint training in the objective on CIFAR-10/CIFAR-100 with ResNet architectures. We also examine the validation performance of the ensemble *and* base learners throughout training on ImageNet. Results are provided in Figure 6. We also provide extensive additional results in Appendix G confirming the generality of this degeneracy. There we consider: larger ensembles, score-averaging, alternative models, additional datasets, and $\overline{\mathrm{ERR}}$ as a metric. Complete experimental details are provided in Appendix B.

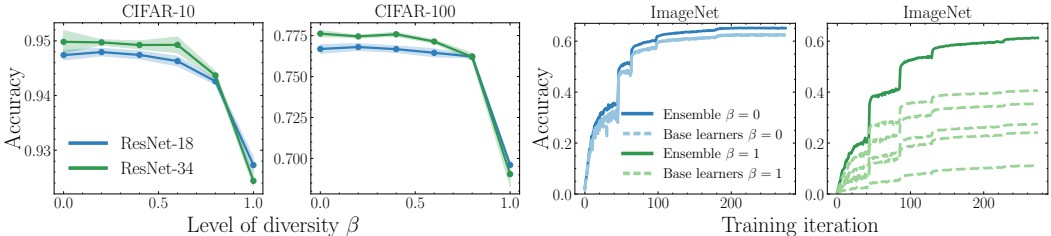

Figure 6: **Practical implications of learner collusion.** Left: Test accuracy with increasing diversity in the objective on CIFAR-10/100 with ResNets. Right: Validation accuracy throughout training ResNet-18 base learners on ImageNet. Jointly trained ensembles ($\beta = 1$) perform significantly worse than independently trained ensembles ($\beta = 0$).

Consistently across all experiments, we find that joint training ($\beta = 1$) results in the worst test set performance. We also note the poor performance among base learners in this regime, highlighting that inflating diversity comes at the cost of degenerate individual performance. Interestingly, across all of our experiments learner collusion typically emerges as $\beta \rightarrow 1$ implying that partial joint training may be a feasible direction for future work. However, in general, independent training emerges as a primary choice of training strategy for deep ensembles.

## 7  Conclusion

In this work, we have provided a detailed account explaining the unexpected observation that jointly trained deep ensembles are generally outperformed by their independently trained counterparts *despite joint training being the objective of interest at test time*. We presented a comprehensive view of diversity in machine learning ensembles through which we analyzed the joint training objective. From this perspective, we uncovered a learner collusion effect in which non-diverse base learners collude to artificially inflate their apparent diversity. Empirically, we both isolated this phenomenon and demonstrated its catastrophic effect on model generalization. Finally, the practical consequences of this limitation were exposed for standard machine learning settings. Note that we also provide a visual summary of this work and its technical contributions in Appendix A. We hope that future work will investigate how to optimize deep ensembles to operate collaboratively whilst avoiding the degeneracies of learner collusion. In Appendix G.7 we include a *negative result* of our initial attempts in this direction. However, several promising avenues remain open (e.g. jointly optimizing worst-case bounds instead [47, 48]).

**Acknowledgments**

We would like to thank Alicia Curth, Fergus Imrie, and the anonymous reviewers for very insightful comments and discussions on earlier drafts of this paper. AJ gratefully acknowledges funding from the Cystic Fibrosis Trust. TL is funded by AstraZeneca. JC is funded by Aviva. This work was supported by Azure sponsorship credits granted by Microsoft's AI for Good Research Lab.

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

# A  Paper Summary

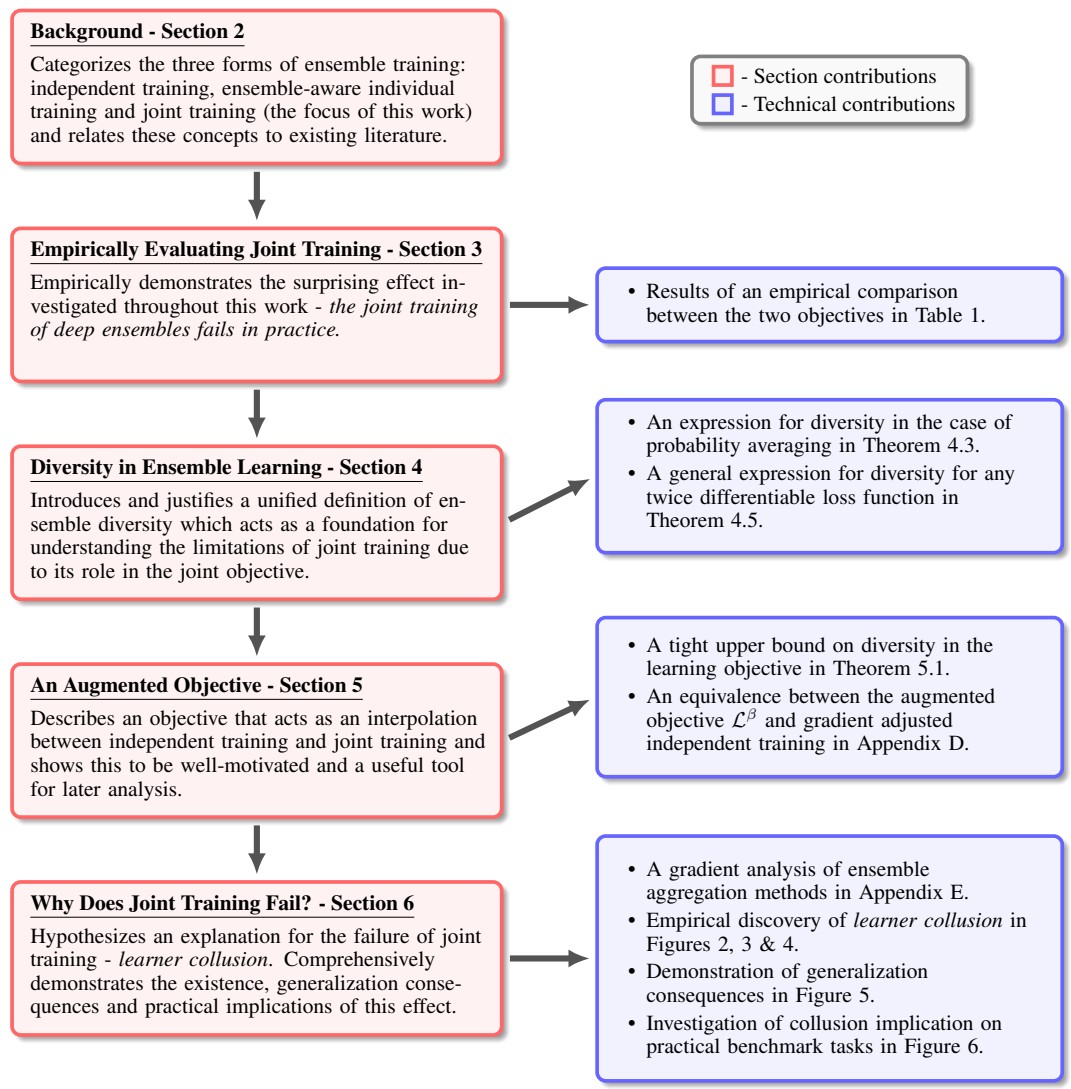

Figure 7: A condensed description of this work and its main contributions.

# B  Experimental Details

**Empirically Evaluating Joint Training (Section 3).** This section compares training using cross-entropy loss as the objective applied to each learner's predictions independently (independent training) and to the ensemble's aggregated predictions (joint training). As described in the main text, we compare the test set performance of these two training methods using various popular architectures on ImageNet [34] evaluating the joint objective on the validation set. Specifically, we use the architectures cited in Table 1 with ensemble size determined by computational limitations. We use implementations from [49] except for MobileViT where the implementation is from [40]. Base learners' are aggregated using their probability outputs (i.e. probability averaging in Section 4.2). We use the standard ImageNet training-validation splits. For all models, we adjusted the following hyperparameters to optimize performance: learning rate, learning rate scheduler (and corresponding decay factor, gamma and frequency), momentum, and weight decay. Batch size was typically set to 128 but was divided by a multiple of 2 for larger base learners. We train models with early stopping and patience of 15 test set evaluations which occur after every 2000 batches. We apply stochastic

gradient descent as our optimizer. We apply standard preprocessing and augmentations consisting of input standardization, random crops, and random horizontal flips.

**Diagnosing Learner Collusion (Section 6.1)**. These experiments investigate the performance of ensembles of multi-layer perceptrons on four regression datasets from the UCI machine learning repository [50]. These are Bioconcentration [51], Weather [52], Facebook [53], and Abalone [54]. Standard preprocessing is applied to each dataset including standardization of the features and targets, one hot encoding of categorical features, and log transformation of skewed features. Due to the smaller size of these tabular datasets, these experiments are less computationally expensive and, therefore, we perform hyperparameter optimization using Bayesian optimization for each run. For each run, we apply 20 iterations of Bayesian optimization searching over the following parameters. Learning rate $\in \{0.01, 0.005, 0.001\}$, batch size $\in \{16, 32, 64\}$, weight decay $\in \{0, 0.01, 0.001\}$, number of base learners $\in \{5, 10, 15, 20\}$, hidden layer neurons $\in \{10, 20, 30\}$, and number of hidden layers $\in \{1, 2, 3\}$. Additionally, the ReLU activation function is used with mean squared error loss. At each run, 20% of the data is randomly used for testing with the remaining data being further split into training and validation with an 80/20 ratio. Each ensemble is trained with early stopping with patience of 5 epochs. The best-performing hyperparameters are selected based on validation set performance and then retrained on the train/validation data for all values of $\beta \in \{0, 0.1, \ldots, 0.9, 1\}$ and evaluated on the test set. This is repeated for 10 runs. The mean $\pm$ a standard deviation is provided in our results.

In Figure 2 we report test set ERR and DIV as MSE and Diversity respectively. In Figure 3 we report the percentage decrease in test set DIV for each value of $\beta$ after the debiasing process described in the main text. In Figure 4, for each model being evaluated on the test set we randomly drop a percentage of the base learners $p \in \{0.2, 0.4, 0.6\}$ and report the mean increase in test set ERR (ensemble mean squared error).

**Learner Collusion Generalizes Poorly (Section 6.2)**. This section uses a similar experimental setting as Section 6.3 on the CIFAR datasets, as described in detail in the next paragraph. The key difference is using Adam optimizer [55] without learning rate decay to evaluate performance smoothly over the course of training. We train for a fixed period of 100 training epochs. We note that this number of training epochs is only for our analysis as the best validation performance generally occurs significantly earlier. We also only use the standard training/testing split as a third split is not required. In Figure 5 we report the difference in ERR measured on the training set vs the test set over the course of training and refer to this metric as the *generalization gap*. This can be expressed as $\mathcal{L}^{\text{test}}(\bar{\mathbf{f}}, \mathbf{y}) - \mathcal{L}^{\text{train}}(\bar{\mathbf{f}}, \mathbf{y})$ where the loss function is cross-entropy loss. Further results from this experiment are provided in Appendix G.

**Practical Implications (Section 6.3)**. In Figure 6 (right) we use the same experimental setup as in Section 3. In the two left-side plots we train and evaluate on CIFAR-10 and CIFAR-100 [56]. We train ensembles of either ResNet-18 or ResNet-34 [35] as base learners with each ensemble consisting of five learners. Base learners are again aggregated using their probability outputs (i.e. probability averaging in Section 4.2). We use the standard training and testing splits with an additional 80/20 split on the training data to provide a validation set in order to ensure our test set performance is independent of implicit bias from the training process. We use a similar learning schedule as proposed in [35]. Specifically, we train for 80 epochs using stochastic gradient descent with momentum of 0.9. The learning rate is initialized at 0.1 and exponentially decays at a rate of 0.2 at epochs $[30, 50, 70]$. We apply weight decay of $5e-4$. Validation performance is evaluated at each epoch with the best-performing model across all 80 epochs evaluated on the test set. A batch size of 128 was used. We repeat each experiment for a total of six runs and report the mean and two standard errors in our results. We apply standard preprocessing and augmentations consisting of input standardization, random crops, random horizontal flips, and random rotations. The key difference in the left figure is that instead of only training deep ensembles using independent training and joint training, we now also consider a hybrid objective using the augmented objective for Section 5. We train this objective for $\beta \in \{0, 0.2, 0.4, 0.6, 0.8, 1\}$ and report accuracy evaluated on the test set in the figure. In the right figure, we provide the validation set accuracy of both ImageNet models throughout training. We provide both the performance of the individual learners and the aggregate ensemble performance.

Additional experiments from this section are provided in Appendix G. Unless otherwise stated, results reported in the appendix are aggregated over three runs.

**Compute**. All experiments are run on Azure NCv3-series VMs powered by NVIDIA Tesla V100 GPUs with 16GB of GPU VRAM.

## C Proofs

### C.1 Optimal Debiasing

We hypothesize that the predictions of each learner in the ensemble is associated with a bias term $b_j$ which cancel in aggregate but significantly affect our measure of diversity. This is the most simple conceivable form of learner collusion and in this section we prove that the debiasing term used in Section C.1 is optimal for this purpose.

We begin by noting that for these biasing terms to not affect the aggregated ensemble prediction we require that $\sum_j (f_j - b_j) = \sum_j f_j$ and therefore we require that $\sum_j b_j = 0$. Then we wish to find the set of such bias terms that minimize the standard diversity, DIV, of the ensemble over the data. Combining these two requirements we arrive at the optimization objective in Equation (3) for an ensemble of size $M$ with $N$ examples.

$$\min_{b_1,\ldots,b_M} \frac{1}{N \cdot M} \sum_{i=1}^{N} \sum_{j=1}^{M} (f_j(\mathbf{x}_i) - b_j - \bar{f}(\mathbf{x}_i))^2 \quad \text{subject to} \quad \sum_{j=1}^{M} b_j = 0. \tag{3}$$

By the method of Lagrange multipliers we obtain $M + 1$ equations for the $M$ values $b_j$ and the Lagrange multiplier $\lambda$

$$\nabla_{b_j} \frac{1}{N \cdot M} \sum_{i=1}^{N} \sum_{j=1}^{M} (f_j(\mathbf{x}_i) - b_j - \bar{f}(\mathbf{x}_i))^2 - \lambda \nabla_{b_j} \sum_{j=1}^{M} b_j = 0 \quad \forall \quad j, \tag{4}$$

$$\sum_{j=1}^{M} b_j = 0. \tag{5}$$

Solving equation 4 for an arbitrary $b_j$, we obtain

$$\lambda = -\frac{2}{N \cdot M} \sum_{i=1}^{N} (f_j(\mathbf{x}_i) - b_j - \bar{f}(\mathbf{x}_i))$$

$$\implies b_j = \frac{\lambda \cdot M}{2} + \frac{1}{N} \sum_{i=1}^{N} (f_j(\mathbf{x}_i) - \bar{f}(\mathbf{x}_i)). \tag{6}$$

And subbing into equation 5

$$\sum_{j=1}^{M} \left( \frac{\lambda \cdot M}{2} + \frac{1}{N} \sum_{i=1}^{N} (f_j(\mathbf{x}_i) - \bar{f}(\mathbf{x}_i)) \right) = 0$$

$$\frac{\lambda \cdot M^2}{2} + \frac{M}{N} \sum_{i=1}^{N} (\bar{f}(\mathbf{x}_i) - \bar{f}(\mathbf{x}_i)) = 0$$

$$\lambda = 0.$$

Finally, we can substitute $\lambda = 0$ back into equation 6 to obtain the solution

$$b_j = \frac{1}{N} \sum_{i=1}^{N} (f_j(\mathbf{x}_i) - \bar{f}(\mathbf{x}_i)) \quad \forall \quad j.$$

## C.2 General Diversity Expression for Multidimensional Outputs

**Theorem 4.5.** *[Diversity for General Loss] For any loss function $\mathcal{L} : \mathcal{Y}^2 \to \mathbb{R}^+$ with $\dim(\mathcal{Y}) \in \mathbb{N}^*$ that is at least twice differentiable, the loss of the ensemble can be decomposed into*

$$\mathrm{DIV} = \frac{1}{2} \sum_{j=1}^{M} w_j \, (\mathbf{f}_j - \bar{\mathbf{f}})^\intercal \cdot \mathcal{H}_f^{\mathcal{L}}(\mathbf{f}_j^*, \mathbf{y}) \cdot (\mathbf{f}_j - \bar{\mathbf{f}})$$

$$\textit{with } \mathbf{f}_j^* = \bar{\mathbf{f}} + c_j(\mathbf{f}_j - \bar{\mathbf{f}}),$$

*where $\mathcal{H}_f^{\mathcal{L}}$ is the Hessian of $\mathcal{L}$ with respect to its first argument and some $c_j \in [0, 1]$ for all $j \in [M]$.*

*Proof.* We follow the same approach taken in [44] but extend their proof to account for a multidimensional output space. For each $j \in [M]$, we define the curve $\gamma_j : [0, 1] \to \mathcal{Y}$ by $\gamma_j(t) = \mathcal{L}[\bar{\mathbf{f}} + t \cdot (\mathbf{f}_j - \bar{\mathbf{f}}), \mathbf{y}]$ for all $t \in [0, 1]$. We note that $\mathcal{L}(\mathbf{f}_j, \mathbf{y}) = \gamma_j(1)$. Since the loss $\mathcal{L}$ is at least twice differentiable, we can perform a first-order Taylor expansion around $t = 0$ with Lagrange remainder:

$$\gamma_j(1) = \mathcal{L}[\bar{\mathbf{f}}, \mathbf{y}] + \nabla_f^\intercal \mathcal{L}[\bar{\mathbf{f}}, \mathbf{y}] \cdot (\mathbf{f}_j - \bar{\mathbf{f}}) + \frac{1}{2}(\mathbf{f}_j - \bar{\mathbf{f}})^\intercal \cdot \mathcal{H}_f^{\mathcal{L}}(\mathbf{f}_j^*, \mathbf{y}) \cdot (\mathbf{f}_j - \bar{\mathbf{f}}),$$

with $\mathbf{f}_j^*$ defined as in the theorem statement. We note that this expression for $\gamma_j(1)$ is *exact* for any twice differentiable loss thanks to the Lagrange's remainder theorem (see e.g. p.878 of [57]). We deduce the following formula for the average loss over the learners:

$$\overline{\mathrm{ERR}} = \sum_{j=1}^{M} w_j \mathcal{L}(\mathbf{f}_j, \mathbf{y})$$

$$= \sum_{j=1}^{M} w_j \gamma_j(1)$$

$$= \underbrace{\mathcal{L}[\bar{\mathbf{f}}, \mathbf{y}]}_{\mathrm{ERR}} + \nabla_f^\intercal \mathcal{L}[\bar{\mathbf{f}}, \mathbf{y}] \cdot \overbrace{\sum_{j=1}^{M} w_j \cdot (\mathbf{f}_j - \bar{\mathbf{f}})}^{0} + \frac{1}{2} \sum_{j=1}^{M} (\mathbf{f}_j - \bar{\mathbf{f}})^\intercal \cdot \mathcal{H}_f^{\mathcal{L}}(\mathbf{f}_j^*, \mathbf{y}) \cdot (\mathbf{f}_j - \bar{\mathbf{f}}),$$

where the second term vanishes due to the identity $\sum_{j=1}^{M} w_j \mathbf{f}_j = \bar{\mathbf{f}}$. Since $\mathrm{DIV} = \overline{\mathrm{ERR}} - \mathrm{ERR}$, this proves the theorem. $\square$

## C.3 Probability Averaged Diversity Term for Classification

**Theorem 4.3.** *For a probability averaged ensemble classifier, trained using cross-entropy loss $\mathcal{L}_{CE}$ with $k^\star \in \mathcal{K}$ denoting the ground truth class, diversity is given by*

$$\mathrm{DIV} = \sum_{j=1}^{M} w_j \mathcal{L}_{CE}(\mathbf{y}, \mathbf{p}_j) - \mathcal{L}_{CE}(\mathbf{y}, \bar{\mathbf{p}}) = \sum_{j=1}^{M} w_j \cdot y(k^\star) \log \frac{\bar{p}(k^\star)}{p_j(k^\star)}$$

*Proof.*

$$\text{DIV} = \sum_{j=1}^{M} w_j \mathcal{L}_{CE}(\mathbf{y}, \mathbf{p}_j) - \mathcal{L}_{CE}(\mathbf{y}, \bar{\mathbf{p}})$$

$$= \sum_{j=1}^{M} w_j D_{KL}(\mathbf{y}||\mathbf{p}_j) + H(\mathbf{y}) - D_{KL}(\mathbf{y}||\bar{\mathbf{p}}) - H(\mathbf{y}) \qquad \text{(where } H \text{ is entropy)}$$

$$= \sum_{j=1}^{M} w_j \sum_{k \in \mathcal{K}} y(k) \log \frac{y(k)}{p_j(k)} - \sum_{k \in \mathcal{K}} y(k) \log \frac{y(k)}{\bar{p}(k)}$$

$$= \sum_{j=1}^{M} w_j \sum_{k \in \mathcal{K}} y(k) \log \frac{\bar{p}(k)}{p_j(k)}$$

$$= \sum_{j=1}^{M} w_j \cdot y(k^\star) \log \frac{\bar{p}(k^\star)}{p_j(k^\star)}$$

$\square$

**Interpretation.** This expression has a sensible interpretation as the diversity of a probability-averaged ensemble. Examining this expression of diversity:

$$\text{DIV} = \sum_{j=1}^{M} w_j \cdot y(k^\star) \log \frac{\bar{p}(k^\star)}{p_j(k^\star)},$$

where $k^\star$ is the true class, and we note that this expression's only non-zero contribution comes from the predictions of the ground-truth class. We now view $p_j(k^\star)$ as coming from a distribution *of probabilities* and assume that $p_j(k^\star)$ is distributed around $\bar{p}(k^\star)$ following a Beta distribution with its mean at $\bar{p}(k^\star)$, i.e. $p_j(k^\star) \sim \texttt{Beta}(\alpha, \beta)$, where $\mathbb{E}[p_j(k^\star)] = \bar{p}(k^\star)$. Consequently, we can examine the expected value of the term with respect to ensemble learners as follows:

$$\mathbb{E}_{p_j(k^\star)}\left[\log \frac{\bar{p}(k^\star)}{p_j(k^\star)}\right] = \mathbb{E}_{p_j(k^\star)}[\log \bar{p}(k^\star)] - \mathbb{E}_{p_j(k^\star)}[\log p_j(k^\star)]$$

$$\approx \mathbb{E}_{p_j(k^\star)}[\log \bar{p}(k^\star)] - \log \mathbb{E}_{p_j(k^\star)}[p_j(k^\star)] + \frac{\mathbb{V}[p_j(k^\star)]}{2\mathbb{E}_{p_j(k^\star)}[p_j(k^\star)]^2}$$

$$= \log \bar{p}(k^\star) - \log \bar{p}(k^\star) + \frac{\mathbb{V}[p_j(k^\star)]}{2\bar{p}(k^\star)^2}$$

$$= \frac{\mathbb{V}[p_j(k^\star)]}{2\bar{p}(k^\star)^2}$$

$$= \frac{\alpha\beta}{(\alpha + \beta)^2(\alpha + \beta + 1)2\bar{p}(k^\star)^2}$$

$$= \frac{\mu(1 - \mu)}{(1 + \phi)2\bar{p}(k^\star)^2}$$

$$= \frac{\bar{p}(k^\star)(1 - \bar{p}(k^\star))}{(1 + \phi)2\bar{p}(k^\star)^2}$$

where the approximation is from taking a second order Taylor expansion around $p_{j_0}(k^\star) = \mathbb{E}_{p_j(k^\star)}[p_j(k^\star)]$ to approximate $\mathbb{E}_{p_j(k^\star)}[\log p_j(k^\star)]$. The fourth line is due to a reparameterization of the Beta distribution in terms of mean $\mu$ and precision $\phi$ parameters with $\alpha = \phi \cdot \mu$ and $\beta = \phi(1 - \mu)$. Using this reparameterization, we arrive at an expression that emits an intuitive interpretation— that for a fixed $\bar{p}$, the smaller the value of $\phi$ (conversely, the larger the variance), the larger the *expected* value of diversity.

Furthermore, we might ask if this expression of diversity in Theorem 4.3 is connected to an information-theoretic quantity as in the score averaging case . While this expression does not take the form of any divergence term that the authors of this work are aware of, the following link can be made.

$$\sum_{j=1}^{M} w_j \sum_{k \in \mathcal{K}} y(k) \log \frac{\bar{p}(k)}{p_j(k)}$$

$$= \sum_{j=1}^{M} w_j \sum_{k \in \mathcal{K}} y(k) \frac{\bar{p}(k)}{\bar{p}(k)} \log \frac{\bar{p}(k)}{p_j(k)}$$

$$= \sum_{j=1}^{M} w_j \mathbb{E}_{\bar{p}(k)} \left[ \frac{y(k)}{\bar{p}(k)} \log \frac{\bar{p}(k)}{p_j(k)} \right]$$

$$= \sum_{j=1}^{M} w_j \mathbb{E}_{\bar{p}(k)} \left[ \gamma(k) \log \frac{\bar{p}(k)}{p_j(k)} \right] \quad \text{where } \gamma(k) = \frac{y(k)}{\bar{p}(k)},$$

$$\rightarrow \sum_{j=1}^{M} w_j D_{KL}(\bar{\mathbf{p}} || \mathbf{p}_j) \quad \text{as } \boldsymbol{\gamma} \rightarrow \mathbf{1}.$$

Therefore, as the ensemble predictive distribution approaches the label distribution, the diversity term approaches the weighted average of the KL-divergence between the ensemble predictive distribution and each of the base learner predictive distributions, analogous to the equivalent expression of the score averaged ensemble.

## C.4    Pathological loss for $\beta > 1$

**Theorem 5.1.** *[Pathological Behavior for $\beta > 1$] We consider a target set $\mathcal{Y} = \mathbb{R}^d$ or $\mathcal{Y} \subset \mathbb{R}^d$ being a compact and convex subset of $\mathbb{R}^d$. Let $\mathcal{L} : \mathcal{Y}^2 \rightarrow \mathbb{R}^+$ be a continuous and finite function on the interior $\text{int}(\mathcal{Y}^2)$. We assume that the loss is not bounded from above on $\mathcal{Y}^2$ in such a way that there exists $\mathbf{y} \in \text{int}(\mathcal{Y})$ and a sequence $(\mathbf{y}_t)_{t \in \mathbb{N}} \subset \mathcal{Y}$ such that $\mathcal{L}(\mathbf{y}_t, \mathbf{y}) \rightarrow +\infty$ for $t \rightarrow \infty$. If $\beta > 1$, then the augmented loss $\mathcal{L}^\beta$ defined in Equation (2) is not bounded from below on $\mathcal{Y}^2$.*

*Proof.* **Case 1.** We first consider the case where $\mathcal{Y} = \mathbb{R}^d$. Without loss of generality, we consider an ensemble such that $w_1, w_2 > 0$. We fix $\mathbf{f}_j = 0 \in \mathcal{Y}$ for all $2 < j \leq M$. We then consider the following sequence of learners: $(\mathbf{f}_{1,t})_{t \in \mathbb{N}} = (\mathbf{y}_t)_{t \in \mathbb{N}}$ and $(\mathbf{f}_{2,t})_{t \in \mathbb{N}} = (-w_1 w_2^{-1} \mathbf{y}_t)_{t \in \mathbb{N}}$. We note that for all $t \in \mathbb{N}$, the ensemble prediction equals zero $\bar{\mathbf{f}}_t = (w_1 \mathbf{y}_t - w_2 w_2^{-1} w_1 \mathbf{y}_t + 0) = 0$. The ensemble loss is therefore constant and finite $\text{ERR}_t \rightarrow \mathcal{L}(0, \mathbf{y}) < +\infty$ as $t \rightarrow \infty$ since $(0, \mathbf{y}) \in \text{int}(\mathcal{Y}^2)$. On the other hand, we have that the average loss diverges as $\overline{\text{ERR}}_t \geq w_1 \mathcal{L}(\mathbf{y}_t, \mathbf{y}) \rightarrow +\infty$ as $t \rightarrow \infty$. We deduce that the augmented loss is not bounded from below when $\beta > 1$ as $\beta \cdot \text{ERR}_t + (1 - \beta) \cdot \overline{\text{ERR}}_t \rightarrow -\infty$ for $t \rightarrow \infty$.

**Case 2.** We now consider the case where $\mathcal{Y} \subset \mathbb{R}^d$ is a compact and convex subset of $\mathbb{R}^d$. Without loss of generality, we consider an ensemble such that $0 < w_1 < 1$. Since $\mathcal{Y}$ is compact, it is closed and admits a topological boundary that we denote $\partial \mathcal{Y}$. Since the loss is finite on the interior $\text{int}(\mathcal{Y}^2)$, our assumption implies that the loss admits a singular point $(\mathbf{y}_\infty, \mathbf{y})$ with $\mathbf{y}_\infty \in \partial \mathcal{Y}$. We fix $\mathbf{f}_j = \mathbf{y}$ for all $1 < j \leq M$. For the first learner, we consider a sequence $(\mathbf{f}_{1,t})_{t \in \mathbb{N}^+}$ with $\mathbf{f}_{1,t} = t^{-1} \cdot \mathbf{y} + (1 - t^{-1}) \cdot \mathbf{y}_\infty$. We trivially have that $\mathbf{f}_{1,t} \rightarrow \mathbf{y}_\infty$ for $t \rightarrow \infty$, so that $\mathcal{L}(\mathbf{f}_{1,t}, \mathbf{y}) \rightarrow \infty$. Hence, once again, we have that the average loss diverges as $\overline{\text{ERR}}_t \geq w_1 \mathcal{L}(\mathbf{f}_{1,t}, \mathbf{y}) \rightarrow +\infty$ as $t \rightarrow \infty$. It remains to show that the ensemble loss ERR is finite. We note that the ensemble prediction also defines a sequence $(\bar{\mathbf{f}}_t)_{t \in \mathbb{N}^+}$ such that for all $t \in \mathbb{N}^+$

$$\bar{\mathbf{f}}_t = w_1 \cdot [t^{-1} \cdot \mathbf{y} + (1 - t^{-1}) \cdot \mathbf{y}_\infty] + \mathbf{y} \cdot \sum_{j=2}^{M} w_j$$

$$= [1 + w_1 \cdot (t^{-1} - 1)] \cdot \mathbf{y} + [w_1 \cdot (1 - t^{-1})] \cdot \mathbf{y}_\infty.$$

We deduce that $\bar{\mathbf{f}}_t \in \mathcal{Y}$ as it is a convex combination of $\mathbf{y}$ and $\mathbf{y}_\infty$. Furthermore, we trivially have that $\bar{\mathbf{f}}_t \to \bar{\mathbf{f}}_\infty = (1 - w_1) \cdot \mathbf{y} + w_1 \cdot \mathbf{y}_\infty$ as $t \to \infty$. We shall now demonstrate that $\bar{\mathbf{f}}_\infty \in \mathrm{int}(\mathcal{Y})$ and conclude. We note that, since $\mathbf{y} \in \mathrm{int}(\mathcal{Y})$, there exists an open ball $B[\mathbf{y}, r]$ of radius $r \in \mathbb{R}^+$ centered at $\mathbf{y}$ such that $B[\mathbf{y}, r] \subseteq \mathcal{Y}$. Let us now consider the open ball $B[\bar{\mathbf{f}}_\infty, (1 - w_1) \cdot r]$. We note that any $\mathbf{y}' \in B[\bar{\mathbf{f}}_\infty, (1 - w_1) \cdot r]$ is of the form

$$\begin{aligned}
\mathbf{y}' &= \bar{\mathbf{f}}_\infty + (1 - w_1) \cdot \mathbf{v} \\
&= (1 - w_1) \cdot (\mathbf{y} + \mathbf{v}) + w_1 \cdot \mathbf{y}_\infty,
\end{aligned}$$

where $\mathbf{v} \in B[0, r]$. Clearly, $(\mathbf{y} + \mathbf{v}) \in B[\mathbf{y}, r] \subseteq \mathcal{Y}$, hence $\mathbf{y}' \in \mathcal{Y}$ as it is a convex combination of two elements of $\mathcal{Y}$. We deduce that $B[\bar{\mathbf{f}}_\infty, (1 - w_1) \cdot r] \subseteq \mathcal{Y}$, which implies that $\mathbf{f}_\infty \in \mathrm{int}(\mathcal{Y})$. We conclude that $\mathrm{ERR}_t \to \mathcal{L}(\bar{\mathbf{f}}_\infty, y)$ by continuity of the loss on $\mathrm{int}(\mathcal{Y}^2)$. We also know that $\mathcal{L}(\bar{\mathbf{f}}_\infty, \mathbf{y})$ is finite as $(\bar{\mathbf{f}}_\infty, \mathbf{y}) \in \mathrm{int}(\mathcal{Y}^2)$. We deduce that the augmented loss is not bounded from below when $\beta > 1$ as $\beta \cdot \mathrm{ERR}_t + (1 - \beta) \cdot \overline{\mathrm{ERR}}_t \to -\infty$ for $t \to \infty$. $\qquad\square$

# D  Gradient Adjusted Targets

Throughout this work we have discussed the two extremes of training neural network ensembles. *Independent training*, where each ensemble member is trained independently and *joint training* where the predictions of the ensemble as a whole is optimized. We have also shown that interpolating between these two extremes via the parameter $\beta$ can result in an effective hybridization. In this section, we provide an alternative interpretation of these intermediate values of $\beta$ as an optimization objective. In particular, we show that the hybrid objective can in fact be interpreted as an independently trained ensemble in which the targets $y$ are adjusted by the gradient of the ensemble.

Specifically, in the case of mean squared error we can adjust the true target $y$ by a step in the opposite direction to the gradient. Then each individual learner $\hat{f}_i$ will have its target adjusted to compensate for the errors of the full ensemble. The size of this step is determined by the parameter $\alpha$. This results in the gradient adjusted target (GAT) objective in Equation (7). Note that throughout this section we consider scalar regression but, of course, these results would hold for multiple outputs by applying the same reasoning to each output.

$$\mathcal{L}^{\mathrm{GAT}}(f_i, \tilde{y}) = \frac{1}{M} \sum_{j=1}^{M} (f_i - \tilde{y})^2, \tag{7}$$

$$\text{where} \quad \tilde{y} = y - \alpha \cdot g$$

$$\text{and} \quad g = \frac{\partial}{\partial \bar{f}} \left( \frac{1}{2} (\bar{f} - y)^2 \right).$$

Conveniently, it can be shown that applying individual training with this GAT objective is equivalent to using the hybrid objective described in equation 2 where the step size $\alpha$ has a similar effect to the interpolation parameter $\beta$.

**Theorem D.1** (Residual adjusted mean squared error). *The ensemble gradient adjusted mean squared error objective*

$$\frac{1}{M} \sum_{i=1}^{M} (f_i - (y - \frac{\alpha}{M} \sum_{j=1}^{M} (f_j - y)))^2, \tag{8}$$

*is proportional to* $\overline{ERR} - (1 - \frac{1}{(\alpha+1)^2}) \cdot DIV$.

*Proof.*

$$\frac{1}{M}\sum_{i=1}^{M}(f_i - (y - \frac{\alpha}{M}\sum_{j=1}^{M}(f_j - y)))^2$$

$$= \frac{1}{M}\sum_{i=1}^{M}(f_i - y + \alpha \cdot \bar{f} - \alpha \cdot y)^2$$

$$= \frac{1}{M}\sum_{i=1}^{M}((f_i - \bar{f}) + (1 + \alpha)(\bar{f} - y))^2$$

$$= \frac{1}{M}\sum_{i=1}^{M}\left[(f_i - \bar{f})^2 + (1 + \alpha)^2(\bar{f} - y)^2 + 2 \cdot (1 + \alpha)(f_i - \bar{f})(\bar{f} - y)\right]$$

$$= \quad\left[\quad \text{DIV} \quad + (1 + \alpha)^2 \underbrace{\text{ERR}}_{=\overline{\text{ERR}} - \text{DIV}} + \quad\quad 0 \quad\quad\right]$$

$$\propto \overline{\text{ERR}} - \left(1 - \frac{1}{(1 + \alpha)^2}\right) \cdot \text{DIV}$$

$\square$

When we take no gradient step at $\alpha = 0$, this is equivalent to $\beta = 0$ or independent training. As we take a larger gradient step we approach joint training noting that as $\alpha \to \infty$ we find $\beta \to 1$. Should we choose to take a step of exactly the gradient itself (i.e. $\alpha = 1$), it is equivalent to hybrid training with $\beta = 0.75$.

Alternatively, we might wish to only consider the $M - 1$ *other* members of the ensemble in our gradient step $g$. In this case we might consider $\bar{f}^{-i} = \frac{1}{M-1}\sum_{j \neq i}^{M-1} f_j$ rather than $\bar{f}$. Again, we find that this can be expressed in terms of $\overline{\text{ERR}}$ and DIV. We begin with a prerequisite in Lemma D.2 and then proceed in Theorem D.3 to show that there is also an exact equivalence in the $M - 1$ case to augmented objective $\mathcal{L}^{\beta}$ described in Section 5.

**Lemma D.2.**

$$\sum_{j}^{M}(\bar{f} - f_j)(y - f_j) = M \cdot \text{DIV}. \tag{9}$$

*Proof.*

$$\sum_{j}^{M}(\bar{f} - f_j)(y - f_j)$$

$$= -\sum_{j}^{M} f_j(\bar{f} - f_j)$$

$$= -M \cdot \bar{f}^2 + \sum_{j}^{M} f_j^2$$

$$= -M \cdot \bar{f}^2 + \sum_{j}^{M}(f_j - \bar{f})^2 + M \cdot \bar{f}^2$$

$$= \sum_{j}^{M}(f_j - \bar{f})^2$$

$$= M \cdot \text{DIV}.$$

$\square$

**Theorem D.3** ($M - 1$ residual adjusted mean squared error). *The $M - 1$ ensemble gradient adjusted mean squared error objective*

$$\frac{1}{M} \sum_{i=1}^{M} (f_i - (y - \frac{\alpha}{M-1} \sum_{j \neq i}^{M} (f_j - y)))^2, \tag{10}$$

*is equivalent to $\overline{\mathrm{ERR}} - \gamma \cdot DIV$ up to a scalar constant where $\gamma = \frac{\alpha}{(\alpha+1)^2} \cdot \frac{M((\alpha+2)M - 2(\alpha+1))}{(M-1)^2}$.*

*Proof.*

$$\frac{1}{M} \sum_{i=1}^{M} (f_i - (y - \frac{\alpha}{M-1} \sum_{j \neq i}^{M} (f_j - y)))^2$$

$$= \frac{1}{M} \sum_{i=1}^{M} (f_i - y + \frac{\alpha}{M-1} \sum_{j \neq i}^{M} f_j - \alpha \cdot y)^2$$

$$= \frac{1}{M} \sum_{i=1}^{M} (\frac{\alpha \cdot M}{M-1} \bar{f} + \frac{M-1-\alpha}{M-1} f_i - (1+\alpha) \cdot y)^2$$

$$= \frac{1}{M} \sum_{i=1}^{M} (\frac{\alpha \cdot M}{M-1} (\bar{f} - f_i) + (1+\alpha) f_i - (1+\alpha) \cdot y)^2 \quad (\text{using: } 1 + \alpha - \frac{\alpha \cdot M}{M-1} = \frac{M-1-\alpha}{M-1})$$

$$= \frac{1}{M} \sum_{i=1}^{M} (\frac{\alpha \cdot M}{M-1} (\bar{f} - f_i) - (1+\alpha) \cdot (y - f_i))^2$$

$$= \left(\frac{\alpha \cdot M}{M-1}\right)^2 \mathrm{DIV} + (1+\alpha)^2 \overline{\mathrm{ERR}} - 2 \cdot \frac{\alpha}{M-1} (1+\alpha) \sum_{i=1}^{M} (\bar{f} - f_i)(y - f_i)$$

$$= \left(\frac{\alpha \cdot M}{M-1}\right)^2 \mathrm{DIV} + (1+\alpha)^2 \overline{\mathrm{ERR}} - 2 \cdot \frac{\alpha}{M-1} (1+\alpha) \cdot M \cdot \mathrm{DIV} \quad (\text{using Lemma D.2})$$

$$= (1+\alpha)^2 \cdot \overline{\mathrm{ERR}} - \frac{\alpha M((\alpha+2)M - 2(\alpha+1))}{(M-1)^2} \cdot \mathrm{DIV}.$$

$\square$

While the scalar $\gamma$ might initially appear unwieldy, at closer inspection it performs a similar function to the full ensemble gradient equivalent in Proposition D.1. The key difference being that this term also accounts for the ensemble size $M$ and ensures that smaller ensembles have a higher weighting on diversity. This relationship is visually depicted in Figure 8 where we again observe a sensible connection to hybrid training at various values of $\beta$.

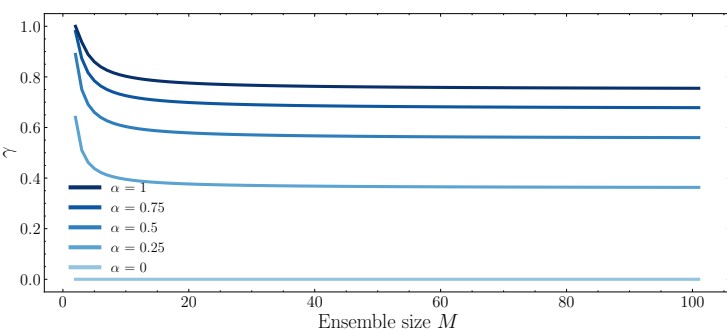

Figure 8: **M − 1 gradient adjusted targets.** $\gamma$ (equivalent to $\beta$ in the augmented objective $\mathcal{L}^\beta$) as a function of ensemble size $M$ and gradient step size $\alpha$. Again we observe a sensible relationship between $\alpha$ and $\gamma$.

**Investigating the relationship of $\beta$ and diversity via GAT.** In the regression experiments in Figure 2 one might intuitively have expected diversity to increase linearly with $\beta$. Instead, we observe a sharp increase in diversity as $\beta \to 1$. The GAT perspective described in this section may provide some insight into why this relationship takes this form. Recall that the $\beta$ term from the augmented objective in Equation (2) is related to the $\alpha$ term the GAT objective in Theorem D.1 according to $\beta = (1 - \frac{1}{(1+\alpha)^2})$. We include a visualization of this relationship in Figure 9. This illustrates that the gradient step $\alpha$ grows exponentially as $\beta \to 1$ resulting in each individual learner having their targets biased excessively (given that the targets are standardized in these experiments) to account for the ensemble errors. This exponential increase in the bias in the targets would appear to be related to the simultaneous increase in diversity at the same point.

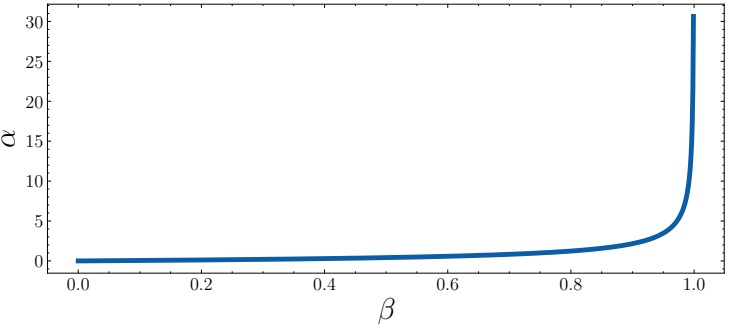

Figure 9: **Understanding the non-linear increase in diversity with $\beta$.** Increasing $\beta$ in our augmented objective is exactly equivalent to individual training with the targets adjusted to account for the errors of the ensemble by a step size of $\alpha$. The relationship between these two terms is given by $\beta = (1 - \frac{1}{(1+\alpha)^2})$ which we visualize in this plot. Thus the gradient step $\alpha$ grows exponentially as $\beta \to 1$ resulting in each individual learner having excessive biases dominate their targets which coincides with an exponential increase in diversity.

# E    Gradient Analysis of Classification Ensemble Averaging Methods

**On the claims of [31].** We begin by taking a closer look at the claims of [31] where, using their notation, they presented the following equation for an equally weighted ensemble using either output averaging or probability averaging.

$$\frac{\partial \ell}{\partial \mathbf{x_i}} = \frac{\partial \ell}{\partial \mu} \frac{\partial \mu}{\partial \mathbf{x_i}} = \frac{\partial \ell}{\partial \mu} \frac{1}{N},$$

where $\mu(\mathbf{x_1}, \cdots, \mathbf{x_N}) = \frac{1}{N} \sum_{i=1}^{N} \mathbf{x_i}$ represents the averaging of a set of $N$ model outputs and $\ell$ is some loss function.

The authors claimed that, because this expression does not depend on $i$, the "gradients backpropagated into all ensemble members are identical". On this basis, they argued that this "lack of gradient diversity" combined with numerical stability issues explains the poor performance of joint training. We wish to make two points on this claim. **(1)** Contrary to what is claimed, this constant gradient effect is only true in the case of score averaging and is not true for probability averaging which also performs poorly under joint training. **(2)** Even in the case of score averaging, although all ensemble members receive the same gradient signal, that signal is a function of the *ensemble performance* under joint training and the *individual performance* under independent training. Therefore the joint training optimal solution would still maximize ensemble performance *if we could obtain the optimal solution*.

We begin with the first point **(1)**. If each $\mathbf{x_i}$ represents the scores of each ensemble member and we presume that the softmax activation $\phi$ is baked into $\ell$, then this is the score-averaged classification setting, and the author's expression holds. However, if we wish to consider the probability averaged setting, then the averaging function should in fact be $\mu(\mathbf{x_1}, \cdots, \mathbf{x_N}) = \frac{1}{N} \sum_{i=1}^{N} \phi(\mathbf{x_i})$ and the derivative takes the following form.

$$\frac{\partial \ell}{\partial \mathbf{x_i}} = \frac{\partial \ell}{\partial \mu} \frac{\partial \mu}{\partial \phi(\mathbf{x_i})} \frac{\partial \phi(\mathbf{x_i})}{\partial \mathbf{x_i}} = \frac{\partial \ell}{\partial \mu} \frac{1}{N} \frac{\partial \phi(\mathbf{x_i})}{\partial \mathbf{x_i}}.$$

It is clear that the final term in this expression is indeed a function of $\mathbf{x_i}$ and therefore each learner can receive unique gradients in the case of probability averaging.

With respect to the second point **(2)**, we can simply note that the loss term $\ell$ is a function of just $\mathbf{x_i}$ in independent training and all $\mathbf{x_1}, \cdots, \mathbf{x_N}$ in joint training. Concretely, we note that the gradient with respect to learner $i$ should be expressed as $\frac{\partial \ell(\mathbf{x_i})}{\partial \mathbf{x_i}}$ for independent training and as $\frac{\partial \ell(\mu(\mathbf{x_1}, \cdots, \mathbf{x_N}))}{\partial \mathbf{x_i}}$ for joint training. This should clarify that although the learners could potentially have constant gradients in the case of joint training, they are still optimizing the correct objective and, therefore, the constant nature of these gradients would not in itself explain why the joint training objective results in poorer ensemble performance. We argue that a better explanation lies in the observation that the additional diversity term in the joint objective surprisingly results in poorer solutions is the observation that these multivariate entangled objectives result in an optimization problem that is significantly more challenging to solve in practice. In other words, individual training performs better as it results in a better solution to a slightly sub-optimal objective.

**A gradient analysis of cross-entropy loss.** Here we extend the previous gradient analysis by comparing the exact gradients obtained for independent training, probability-averaged joint training and score-averaged joint training in the case of cross-entropy loss. We use the following notation. $f_{j,k}$ is the pre-activation output, or score, for learner $j$ of class $k$. $\phi$ is the softmax function with $\phi(f_j)_k$ referencing the the $k$'th element of the softmax function applied to the vector $f_j$. $\mathcal{L}$ is the loss function.

We summarize the objectives for the three cases in Table 2. The columns correspond to the training method (independent vs joint) paired with the output averaging method (probability vs score). The first row corresponds to a general loss function, while the others are the specific cases of cross-entropy loss. The third row is simply re-expressing the second row by considering the true class $k^\star$ such that $y_{k^\star} = 1$.

Table 2: **Ensemble loss functions**. A summary of primary loss functions used throughout this work. The first row corresponds to the expressions for a general loss function, while the following rows contain equivalent expressions for cross-entropy loss. Note that the final row is just expressing the previous row with the correct class $k^\star$ replacing the sum.

|  | Independent training | Joint training | |
|---|---|---|---|
| **Output aggregation** | **Both** | **Probability averaging** | **Score averaging** |
| General loss | $\mathcal{L}^{\text{ind}} = \frac{1}{M} \sum_j \mathcal{L}(\phi(f_j), y)$ | $\mathcal{L}^{\text{pa}} = \mathcal{L}(\frac{1}{M} \sum_j \phi(f_j), y)$ | $\mathcal{L}^{\text{sa}} = \mathcal{L}(\phi(\frac{1}{M} \sum_j f_j), y)$ |
| Cross-entropy loss | $-\frac{1}{M} \sum_k \sum_j y_k \log(\phi(f_j)_k)$ | $-\sum_k y_k \log\left(\frac{1}{M} \sum_j \phi(f_j)_k\right)$ | $-\sum_k y_k \log\left(\phi(\frac{1}{M} \sum_j f_j)_k\right)$ |
| Cross-entropy loss $(k = k^\star)$ | $-\frac{1}{M} \sum_j \log(\phi(f_j)_{k^\star})$ | $-\log\left(\frac{1}{M} \sum_j \phi(f_j)_{k^\star}\right)$ | $-\log\left(\phi(\frac{1}{M} \sum_j f_j)_{k^\star}\right)$ |

Now we evaluate the gradient of each training method and output averaging combination with respect to an arbitrary learner $i$ for an arbitrary class $k$. We begin with the case of independent training.

$$\frac{\partial \mathcal{L}^{\text{ind}}}{\partial f_{i,k}} = -\frac{1}{M} \frac{\partial}{\partial f_{i,k}} \sum_j \log(\phi(f_j)_{k^\star}) = -\frac{1}{M} \frac{\frac{\partial}{\partial f_{i,k}} \phi(f_i)_{k^\star}}{\phi(f_i)_{k^\star}}$$

$$= \begin{cases} -\frac{1}{M} \cdot (1 - \phi(f_i)_{k^\star}) & \text{if } k = k^*, \\ \frac{1}{M} \cdot \phi(f_i)_k & \text{if } k \neq k^*. \end{cases}$$

Next we examine the equivalent gradients of the joint training objective with probability averaged predictions.

$$\frac{\partial \mathcal{L}^{\text{pa}}}{\partial f_{i,k}} = -\frac{\partial}{\partial f_{i,k}} \log \left( \frac{1}{M} \sum_j \phi(f_j)_{k^\star} \right) = -\frac{\frac{\partial}{\partial f_{i,k}} \phi(f_i)_{k^\star}}{\sum_j \phi(f_j)_{k^\star}}$$

$$= \begin{cases} -\frac{\phi(f_i)_{k^\star}(1-\phi(f_i)_{k^\star})}{\sum_j \phi(f_j)_{k^\star}} & \text{if } k = k^*, \\ \frac{\phi(f_i)_k \phi(f_i)_{k^\star}}{\sum_j \phi(f_j)_{k^\star}} & \text{if } k \neq k^*. \end{cases}$$

Finally, we calculate the same gradients for the case of score averaging.

$$\frac{\partial \mathcal{L}^{\text{sa}}}{\partial f_{i,k}} = -\frac{\partial}{\partial f_{i,k}} \log \left( \phi(\frac{1}{M} \sum_j f_j)_{k^\star} \right) = -\frac{\frac{\partial}{\partial f_{i,k}} \phi(\frac{1}{M} \sum_j f_j)_{k^\star}}{\phi(\frac{1}{M} \sum_j f_j)_{k^\star}}$$

$$= \begin{cases} -\frac{1}{M} \cdot \left( 1 - \phi(\frac{1}{M} \sum_j f_j)_{k^\star} \right) & \text{if } k = k^*, \\ \frac{1}{M} \cdot \phi(\frac{1}{M} \sum_j f_j)_k & \text{if } k \neq k^*. \end{cases}$$

From these gradients, we can draw the following conclusions for a given ensemble member $f_i$.

- Independent training - Each member receives a variable gradient that is a function of their own predictions but not the predictions of the rest of the ensemble. The gradient for learner $f_i$ takes the form $g(f_i)$.

- Joint training with probability averaging - Each member receives a variable gradient that is a function of their own predictions and the predictions of the other ensemble members. The gradient for learner $f_i$ takes the form $g(i, f_1, \cdots, f_M)$.

- Joint training with score averaging - Each member receives a constant gradient that is a function of their own predictions and the predictions of the other ensemble members. The gradient for learner $f_i$ takes the form $g(\bar{f})$.

## F   Learner Collusion in Regression

In this section, we provide an illustrative example of the learner collusion effect in the case of regression with mean squared error. We begin by recalling the relevant decomposition from Equation (1) for an equally weighted ensemble.

$$\text{ERR} = \overline{\text{ERR}} - \text{DIV},$$

$$(\bar{f} - y)^2 = \frac{1}{M} \sum_j (f_j - y)^2 - \frac{1}{M} \sum_j (f_j - \bar{f})^2. \tag{11}$$

Without loss of generality, we consider each base learner to predict some ensemble prediction $f \in \mathbb{R}$ and a learner-dependent scalar adjustment $c_j \in \mathbb{R}$ such that $f_j = f + c_j$. We also write $\frac{1}{M} \sum_j c_j = \bar{c}$. It is straightforward to check that Equation (11) can then be expressed as

$$\text{ERR} = \frac{1}{M} \sum_j (f - y + c_j)^2 - \frac{1}{M} \sum_j (c_j - \bar{c})^2,$$

$$= \underbrace{(f - y)^2 + \frac{1}{M} \sum_j c_j^2 + \frac{1}{M} \sum_j c_j(f - y)}_{\overline{\text{ERR}}} - \underbrace{\frac{1}{M} \sum_j (c_j - \bar{c})^2}_{\text{DIV}}.$$

Setting aside $f$, we focus on the optimization of the $c_j$'s. We note that in the setting of independent training where we only minimize $\overline{\text{ERR}}$, the unique minimum is achieved when $c_j = 0 \quad \forall \quad j$. However, when we add the additional DIV term, this changes. When optimizing the joint objective, ERR, *any* solution satisfying $\bar{c} = y - f$ will minimize this loss. This is exactly learner collusion. We

note that as we optimize on training data, loss typically approaches zero, and $\bar{c} \to 0$. Furthermore, if we were to require that the two terms $\overline{\text{ERR}}$ and DIV should be relatively, approximately equal, we note that this would only occur as the $c_j$ terms become large. Specifically, $\overline{\text{ERR}} \approx \text{DIV}$ as $|c_j| \to \infty$. While this analysis illustrates how artificial diversity may emerge in the regression setting, we note that this is still not *guaranteed* to result in poor test performance in all cases. One such example is the Facebook dataset in Figure 2 where the test mean squared error is largely unaffected by this excessive diversity. In this case, it appears that almost the entire predictive is contained within a single feature resulting in no benefit from true diversity and, therefore, no substantial impact due to learner collusion

Finally, we highlight that during independent training, the learners are not aware of each other's predictions and, therefore, the premise that there would be a single shared ensemble prediction can not hold. In this case, all diversity will be due to external factors such as random initialization and improved performance over a single model can be explained by standard ensemble theory [e.g. 1, 2]. In the case of joint training, the premise that the ensemble produces a single prediction disguised as multiple predictions (i.e. $f_j = \hat{f} + c_j$ where $\frac{1}{M} \sum_j c_j = 0$) is plausible and even likely. However, if this is the case, then clearly there is no genuine diversity among the learners implying that overfitting and poor generalization may become more likely.

## G    Additional Results

### G.1    Generalization Gap Loss Plots

This section provides an extended analysis of the experiments investigating the generalization consequences of learner collusion discussed in Section 6.2. Figure 10 provides the same generalization gap plots from Figure 5, but decomposed into the training and testing loss. For analysis purposes, we trained these models for 100 epochs, but this plot highlights that convergence occurs much earlier. These results reveal that the generalization gap is caused by both a slower training set convergence in addition to a greater test set loss. This slower convergence may in part be due to the two terms of the joint objective ($\beta = 1$) competing against each other in the direction of their gradient updates. More concerning is the larger test set loss. This indicates that the solution achieved as training loss eventually becomes low, generalizes much more poorly than the solution achieved by the independent training objective ($\beta = 0$).

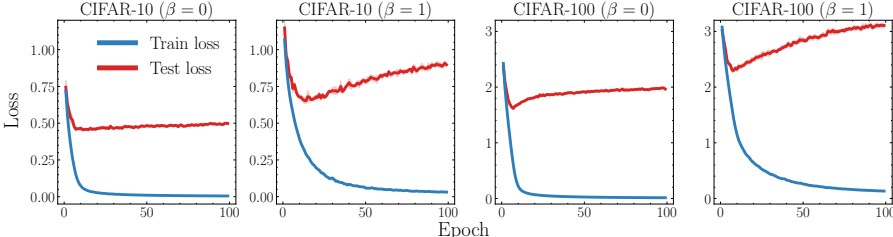

Figure 10: **Train & test loss.** The jointly trained model ($\beta = 1$) converges more slowly on the training set and reaches a solution that generalizes more poorly than the independent objective ($\beta = 0$).

We also provide a decomposition of the loss into individual error ($\overline{\text{ERR}}$) and diversity (DIV) on both the training set (Figure 11) and the test set (Figure 12). From the training plots, it appears that the joint objective ($\beta = 1$) maximizes diversity early in training before beginning to minimize individual error later. This aligns with the learner collusion effect we describe in Section 6.1 as it is trivial to inflate diversity while minimizing the individual error requires solving the true objective of the task. The test set decomposition in Figure 12 reinforces that the solutions obtained with joint training generalize poorly beyond the training data.

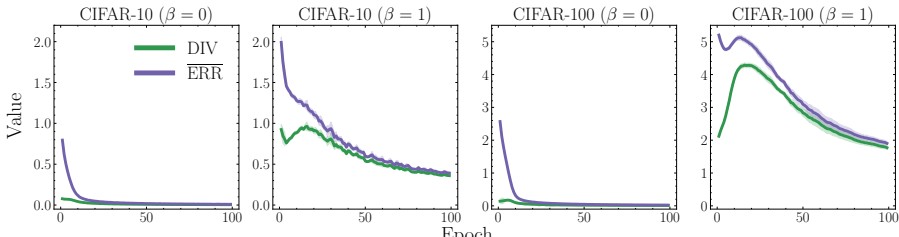

Figure 11: **Train loss decomposition.** The training loss decomposed into diversity (DIV) and individual error ($\overline{\text{ERR}}$) highlights that diversity is trivially inflated early in the training process when joint training is applied ($\beta = 1$).

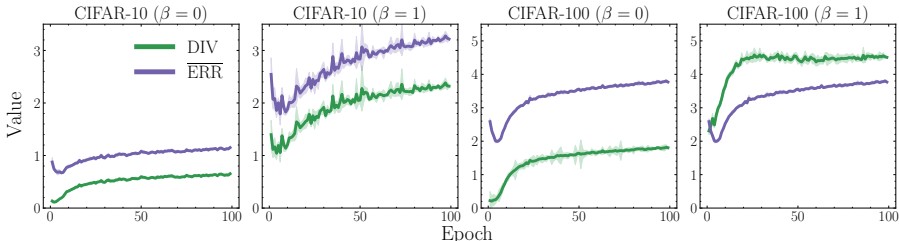

Figure 12: **Test loss decomposition.** The test loss decomposed into diversity (DIV) and individual error ($\overline{\text{ERR}}$) reinforces that optimizing for learner collusion does not generalize effectively onto a test set.

## G.2 Larger Ensembles

In Figure 13 we reproduce the experiments with ResNet-18 base learners from Figure 6 (left) but with a larger ensemble such that $M = 10$. We observe that, while there is a small improvement in test accuracy, the shape of the curves with increasing $\beta$ are consistent with those of the smaller ensemble.

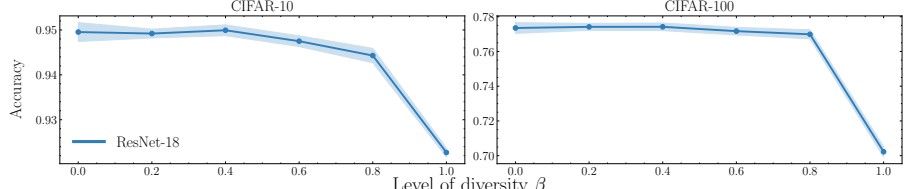

Figure 13: **Larger ensembles.** Identical experiments with larger ensembles ($M = 10$) result in a similar trend in the accuracy curves.

## G.3 Score Averaging

Figure 14 includes the results of repeating the experiments in the top row of Figure 6 using score-averaging rather than probability averaging as described in Section 4.2. Similarly to previous works, we find that joint training performs poorly in this setting too. We also note that test set performance is similar to that of probability averaging on both datasets.

## G.4 CNN Ensembles

We repeat the experiments described in Section 6.3 with ensembles consisting of base learners with a vanilla convolutional neural network architecture. The architecture of each of these base learners is provided in Table 3. Dropout with probability 0.1 is applied throughout the network during training. Furthermore, we apply Adam optimizer [55] with learning rate 0.001 and early stopping

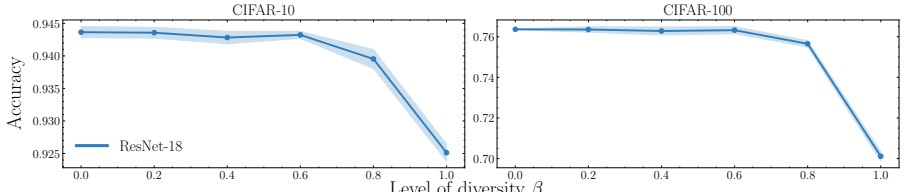

Figure 14: **Score averaging.** This objective also performs significantly worse when applying joint training ($\beta = 1$).

with patience of 5 epochs. Otherwise, all experimental parameters are consistent with those previous experiments. We apply this ensemble to CIFAR-10 with ensembles of size $M = 5$ and $M = 10$ with results provided in Figure 15. The results of this experiment are consistent with those of previous architectures and we again note the poor test set performance of the jointly trained model ($\beta = 1$). Interestingly, we find in both cases that the lowest test loss is achieved at interpolating values of $\beta$ as achieved by the augmented objective $\mathcal{L}^\beta$ as described in Section 5.

Table 3: Vanilla Convolutional Neural Network Architecture.

| Layer Type | Hyperparameters | Activation Function |
|---|---|---|
| Conv2d | Input Channels:3 ; Output Channels:10 ; Kernel Size:5 ; Stride:2 ; Padding:1 | ReLU |
| Conv2d | Input Channels:10 ; Output Channels:20 ; Kernel Size:5 ; Stride:2 ; Padding:1 | ReLU |
| Flatten | Start Dimension:1 | |
| Linear | Input Dimension: 500 ; Output Dimension: 50 | ReLU |
| Linear | Input Dimension: 50 ; Output Dimension: 10 | |

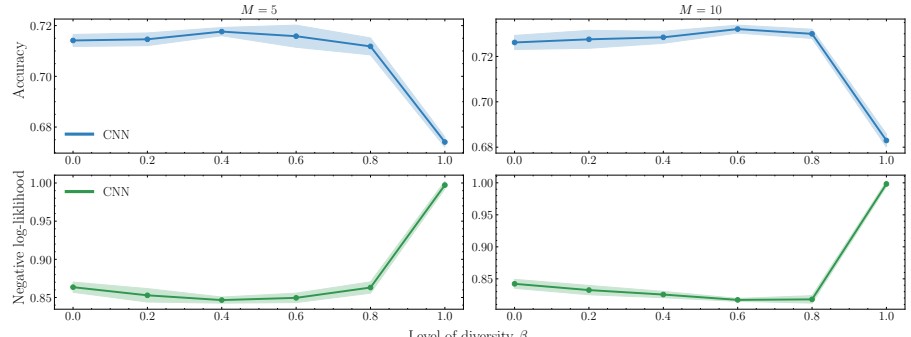

Figure 15: **CNN ensembles.** Standard architectures achieve similarly poor results for joint training ($\beta = 1$) but obtain their lowest test loss at intermediate values ($\beta \in (0, 1)$) using $\mathcal{L}^\beta$.

### G.5 SVHN with VGG architecture

In this experiment we repeat the experiments in Section 6.3 on the SVHN dataset [58] using a VGG style architecture [14]. We report the exact architecture in Table 4. Again, we use ensembles of size 5. We use the same optimization protocol as applied to the ResNet-based ensembles as reported in Appendix B. Consistent with all previous experiments we observe that joint training again results in degenerate performance in this setting.

### G.6 $\overline{\text{ERR}}$ as a metric

For completeness we report the $\overline{\text{ERR}}$ test set performance corresponding to the experiments in Figure 6 (left). We note that this is exactly the ensemble loss which is optimized at $\beta = 1$. This is useful to report as it is the exact quantity being optimized during training. Consistent with all previous results, performance is worst as $\beta \to 1$ (i.e. as we approach optimizing for the metric being reported).

Table 4: VGG Architecture.

| Layer Type | Hyperparameters | Activation Function | Dropout rate |
|---|---|---|---|
| Conv2d | Input Channels:3 ; Output Channels:64 ; Kernel Size:3 ; Stride:1 ; Padding:1 | BN & ReLU | 0.3 |
| Conv2d | Input Channels:64 ; Output Channels:64 ; Kernel Size:3 ; Stride:1 ; Padding:1 | BN & ReLU | 0 |
| MaxPool2d | Kernel Size:2 ; Padding:2 | | |
| Conv2d | Input Channels:64 ; Output Channels:128 ; Kernel Size:3 ; Stride:1 ; Padding:1 | BN & ReLU | 0.4 |
| Conv2d | Input Channels:128 ; Output Channels:128 ; Kernel Size:3 ; Stride:1 ; Padding:1 | BN & ReLU | 0 |
| MaxPool2d | Kernel Size:2 ; Padding:2 | | |
| Conv2d | Input Channels:128 ; Output Channels:256 ; Kernel Size:3 ; Stride:1 ; Padding:1 | BN & ReLU | 0.4 |
| Conv2d | Input Channels:256 ; Output Channels:256 ; Kernel Size:3 ; Stride:1 ; Padding:1 | BN & ReLU | 0.4 |
| Conv2d | Input Channels:256 ; Output Channels:256 ; Kernel Size:3 ; Stride:1 ; Padding:1 | BN & ReLU | 0 |
| MaxPool2d | Kernel Size:2 ; Padding:2 | | |
| Conv2d | Input Channels:256 ; Output Channels:512 ; Kernel Size:3 ; Stride:1 ; Padding:1 | BN & ReLU | 0.4 |
| Conv2d | Input Channels:512 ; Output Channels:512 ; Kernel Size:3 ; Stride:1 ; Padding:1 | BN & ReLU | 0.4 |
| Conv2d | Input Channels:512 ; Output Channels:512 ; Kernel Size:3 ; Stride:1 ; Padding:1 | BN & ReLU | 0 |
| MaxPool2d | Kernel Size:2 ; Padding:2 | | |
| Conv2d | Input Channels:512 ; Output Channels:512 ; Kernel Size:3 ; Stride:1 ; Padding:1 | BN & ReLU | 0.4 |
| Conv2d | Input Channels:512 ; Output Channels:512 ; Kernel Size:3 ; Stride:1 ; Padding:1 | BN & ReLU | 0.4 |
| Conv2d | Input Channels:512 ; Output Channels:512 ; Kernel Size:3 ; Stride:1 ; Padding:1 | BN & ReLU | 0 |
| MaxPool2d | Kernel Size:2 ; Padding:2 | NA | |
| Flatten | Start Dimension:1 | | 0.5 |
| Linear | Input Dimension: 512 ; Output Dimension: 512 | BN & ReLU | 0.5 |
| Linear | Input Dimension: 512 ; Output Dimension: 10 | | 0 |

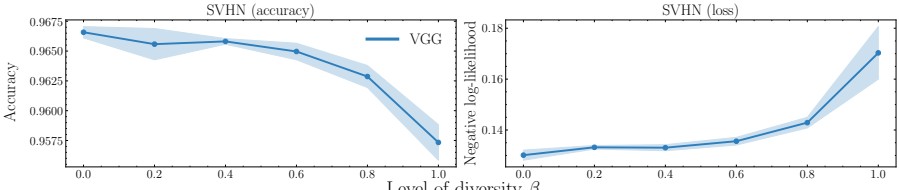

Figure 16: **VGG ensembles on SVHN data.** The same phenomenon emerges when we consider an alternative VGG style architecture on the SVHN dataset.

## G.7 Learner dropout as a resolution

In this section we provide a negative result in which we attempt to resolve learner collusion by randomly removing a subset of base learners from the ensemble at each batch during training. The hypothesis is that if we drop a sufficiently large portion of base learners but still perform joint training on the remainder, this should remove the ability to deterministically collude and, therefore, cause inflated diversity to be actively harmful even on the training data. One might hope that this would cause the ensemble to avoid this degeneracy. In what follows we provide the details of this experiment.

We repeat the setup on CIFAR-10 with ResNet-18 architecture as described earlier in the paper. We train each model using the joint objective, but at each batch we drop a proportion $p \in [0, 0.2, 0.4, 0.6]$ of randomly selected learners from the ensemble. We then investigate if this (a) reduces collusion and (b) improves ensemble performance. The results of this experiment are included in Figure 18. We find that dropping learners does indeed significantly reduce the diversity score indicating a reduction in learner collusion (although a reasonably large proportion of learners is required to be dropped). Unfortunately, the improvement in performance by reducing collusion is negated by a decrease in individual performance due to the base learners becoming weaker on average. The reason why is apparent once we notice that by dropping a base learner from some proportion of batches, this is exactly equivalent to bootstrapping. While bootstrapping has been effective for ensembles of weak learners (e.g. random forest) it has already been shown to be harmful for deep ensem-

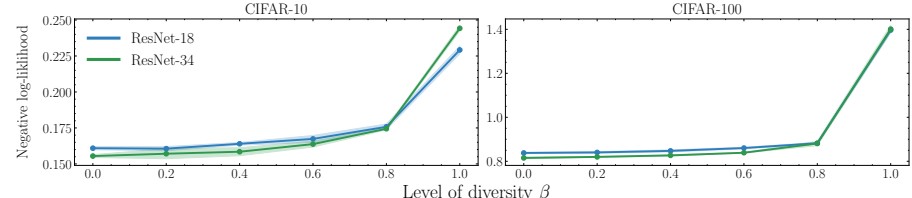

Figure 17: **Loss plots.** Corresponding ensemble loss ($\overline{\text{ERR}}$) for the results presented in Figure 6 (left).

bles [12]. Unfortunately, this resolution appears to only reduce the effect of learner collusion by simultaneously harming the performance of the ensemble due to bootstrapping.

Proposing methods to overcome learner collusion is a valuable but non-trivial direction for future work that we and, we hope, others in the research community will pursue. We believe that the comprehensive diagnosis of the issue we have presented in this work will provide a foundation for future methodological progress.

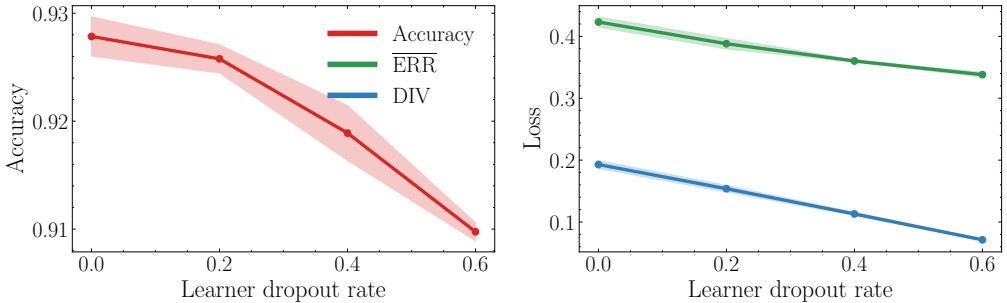

Figure 18: **Investigating learner dropout.** We evaluate randomly dropping a subset of learners during training to remedy the learner collusion effect. We repeat the setup on CIFAR-10 with ResNet-18 architecture from Figure 6 with $\beta = 1$ but at each training batch, we drop a proportion of learners in $[0.0, 0.2, 0.4, 0.6]$. We find that, while this seemingly does reduce learner collusion, the benefit is negated by a decrease in individual performance due to base learners becoming weaker on average. Results are reported over 5 runs.

