# OpenReview forum: "Joint Training of Deep Ensembles Fails Due to Learner Collusion"
_NeurIPS.cc/2023/Conference — NeurIPS 2023 poster_

### Official Review · Reviewer_a4xj · 2023-06-22

**Soundness:** 3 good
**Presentation:** 4 excellent
**Contribution:** 3 good
**Rating:** 7
**Confidence:** 3

**Summary:**

The authors aim to answer why joint training of ensembles does not work as well as separately training the members of the ensembles, before ensembling them. This is a well-known empirical fact, but the authors attempt to give a theoretical understanding of it, which is novel to my knowledge.

To reach their objective, they consider an expression for diversity (DIV) for probability averaging, which they motivate thru a (known) regression example. Then they generalize that expression to any twice differentiable loss function.

The authors study DIV in more depth by (upper) bounding it, then using it as a tunable regularization tool to interpolate between joint training and separate training of ensembles.

Then the authors hypothesize that jointly training the ensemble may artificially bump up DIV without yielding proper generalization. They say that this happens because of "learner collusion" a phenomenon where two learners are displaced artificially by a constant bias term without changing the overall performance of the ensemble. They devise empirical measures to test that hypothesis (by studying generalization gaps, diversity explosion, learner codependence etc.)





**Strengths:**

S1: The paper is really well written. Thanks to the authors for building my reading experience so well.

S2: The explanation of the phenomenon (to my knowledge) is novel, and gives me a new way of thinking about the well-known failure case of jointly training ensembles.

S3: It is great that the authors have theory that is paralleled with experiments on real-world data.

**Weaknesses:**

W1: Not sure I understand what happens when we vary $\beta$. I think the paper can improve in a more detailed discussion about a more thorough study of the effect of collusion in the intermediate values of $\beta$. See my Q2 in the questions below.

W2: Typically, we expect to be able to devise better models after we have a better understanding of an underside phenomenon. The paper lacks a discussion about this. It's fine to leave it as future work, but it would be useful to discuss some directions a bit.

**Questions:**

Q1: In Figure 2, why does Facebook seem to be different from the overall trend?

Q2: If we get $\beta \approx 1$, then great, we have increased diversity and this gives us some hints as to why joint training might fail. However, as we vary $\beta$, I do not gain any intuition. Is the collusion effect a monotonic one? It seems so from Figure 6. If there is some $\beta > 0$ then do we always have that effect? Shouldn't intuition tell us that there should be a "sweet spot" for $\beta$? If not, then why would we be interested in training learners jointly to any extent? For example, could we do something more explicit to prevent the learner's collusion?

**Limitations:**

The authors have adequately addressed the limitations.

---

> ### Author Rebuttal · Authors · 2023-08-09
>
> We thank this reviewer for their constructive feedback. We have addressed the points raised below which have helped us to further improve our submissions clarity and contribution.
>
>   $~$
>
> * _Further discussion on the effects of varying $\beta$_ - The non-linear relationship between $\beta$ and the mean squared error is indeed a notable phenomenon that we only addressed in the appendix. Let us provide a more succinct discussion on this point.
>
>   $~$
>
>   Figures 2-4 do highlight a sudden jump in learner collusion as $\beta \to 1$ in the case of regression rather than a gradual increase.  Why does this occur? A useful explanation lies in the gradient-adjusted target (GAT) analysis in App D. To briefly summarize, this section considers applying individual training but with an adjusted target which is a function of the gradient of the ensemble error. Specifically, we adjust the targets to account for the errors of the ensemble. Formally, we optimize $\frac{1}{M}\sum_{j=1}^M (f_j - \bar{y})^2$ where $\bar{y}$ is the adjusted target given by $\bar{y} = y - \alpha \cdot g$. Here $g$ is the ensemble loss gradient and $\alpha$ is a step size parameter determining how big a step the target of an individual learner should take in order to account for the errors of the ensemble. In Thm. D.1 we show that this GAT objective is exactly equivalent to our augmented objective in Sec. 5 with $\beta = (1 - \frac{1}{(1 + \alpha)^2})$. Given this relationship, we might notice that the gradient step $\alpha$ grows exponentially as $\beta \to 1$ resulting in each individual learner having their targets biased excessively to account for the ensemble errors (**please see the new figure in the attached PDF visualizing this relationship**). Given this perspective, it is unsurprising that learner collusion does not grow linearly with $\beta$.
>
>   $~$
>
>   The reviewer's question regarding a "sweet spot" of $\beta$ for optimizing diversity is an interesting one. Considering that joint training only occurs at exactly $\beta = 1$ we had the same intuition that low values of $\beta$ may still be beneficial. Interpolating values of $\beta$ sometimes appear to achieve the best performance (see e.g. ResNet-18 in Fig. 6 left and CNN in Fig. 14) but often this is not the case (e.g. Fig. 15). We hope that, building upon our findings, future work will be able to provide methods for optimizing for higher levels of diversity whilst avoiding learner collusion and thereby enable us to study the effects of increasing diversity _without_ this obscuring degeneracy.
>
>   $~$
>
>   **Action taken**: We have added a succinct summary of this discussion to the main text. We have also added this new plot and refined the discussion in App D.
>
>   $~$
>
> * _Discussion in improved methods based on our findings_ - We agree that our work could benefit from some additional discussion on future methodological work building upon ours. This is something we are happy to expand upon in our conclusion. Reviewer pZUZ suggested randomly dropping a subset of base learners during training. The hypothesis being that if we drop a sufficiently large portion of base learners in, say, each batch but still perform joint training on the remainder, this should cause inflated diversity to be actively harmful even on the training data and therefore force the ensemble to avoid this degeneracy. We decided to try out this idea which we describe next.
>
>   $~$
>
>   We repeat the setup on CIFAR-10 with ResNet-18 architecture as described in the paper. We train each model using the joint objective, but at each batch we drop a proportion $p \in [0, 0.2, 0.4, 0.6]$ of randomly selected learners from the ensemble. We then investigate if this (a) reduces collusion and (b) improves ensemble performance. **The results of this experiment are included in the attached pdf**. We find that dropping learners does indeed significantly reduce the diversity score indicating a reduction in learner collusion (although a reasonably large proportion of learners is required to be dropped). Unfortunately, the improvement in performance by reducing collusion is negated by a decrease in individual performance due to the base learners becoming weaker on average. The reason why is apparent once we notice that by dropping a base learner from some proportion of batches, this is exactly equivalent to bootstrapping. While bootstrapping has been effective for ensembles of weak learners (e.g. random forest) it has already been shown to be harmful for deep ensembles [1]. Unfortunately, despite being a sensible idea at the outset, this resolution can only reduce the effect of learner collusion by simultaneously harming the performance of the ensemble due to bootstrapping.
>
>   $~$
>
>   Proposing methods to overcome learner collusion is a valuable but non-trivial direction for future work that we and, we hope, others in the research community will pursue. We believe that the comprehensive diagnosis of the issue we have presented in this work will provide a foundation for future methodological progress.
>
>   $~$
>
>   **Action taken**: We have added the results of this experiment as a negative result in order to aid future research and expanded upon our discussion of future work in the conclusion.
>
>   $~$
>
>   [1] Nixon, Jeremy, Balaji Lakshminarayanan, and Dustin Tran. "Why are bootstrapped deep ensembles not better?." ''I Can't Believe It's Not Better!''NeurIPS 2020 workshop. 2020.
>
>   $~$
>
> * _Facebook dataset_ - Please see our response in the general comment section.

---

> > ### Comment · Senior_Area_Chairs · 2023-08-18
> > **@a4xj: Please engage with rebuttal**
> >
> > The authors have posted their rebuttal to your review—does it affect your opinion? It'd be helpful if you can at least acknowledge having read the rebuttal, even if you don't find it convincing. Thanks!

---

> > ### Comment · Reviewer_a4xj · 2023-08-19
> > **Great rebuttal.**
> >
> > Thank you for your clarifications. I read the discussions with the other reviewers too. I would like to see the paper accepted.

---

### Official Review · Reviewer_pZUZ · 2023-07-04

**Soundness:** 3 good
**Presentation:** 3 good
**Contribution:** 3 good
**Rating:** 7
**Confidence:** 3

**Summary:**

Ensembles are a simple but powerful way to improve model performance. Typically, ensembles are used to improve performance by training each model independently and then using them jointly. However, unlike previous ML methods that require ensemble members to be trained individually, Deep Ensemble, an ensemble of deep learning models, can also be trained jointly. Considering that joint performance is the true objective of Deep Ensemble, it seems more natural to train jointly rather than individually. However, in practice, joint training leads to poor performance and poor generalization. The authors show theoretically and empirically that this is due to ensemble diversity. From this, they propose an augmented objective $\mathcal{L}^\beta$ that allows for a linear interpolation between joint and independent training. Finally, they hypothesize that this is due to learner collusion, a phenomenon where each model has a bias when jointly training an ensemble, and show through experiments that this is indeed the case.

**Strengths:**

- The notation is well organized and the formulas are easy to follow.
- The problems covered in previous studies are well summarized and theoretically explained in Sections 2, 4, and 5.
- The reason why joint training of Deep Ensemble fails is effectively explained through *learner collusion* and is well supported experimentally.

**Weaknesses:**

- The authors do not propose a way to prevent learner collusion while doing joint training, so there is not much contribution in terms of practicality. The possibility of partial joint training by weighting between independent training and joint training using an augmented objective is presented as a future work, which is already analyzed in various ways in Webb et al. (2020).
- (Minor) The overall formatting of the paper is less polished and a bit difficult to read.
    - Figures are scattered throughout the page. If possible, the figure should be located at the top of the page.
    - The text in the figure is too small to read the results. Increase the size of the text so that it is not too different from the size of the main text.
    - The lack of titles in the references makes it very difficult to identify the relevant papers. **Please fix this.**
-----
(Webb et al., 2020) [To Ensemble or Not Ensemble: When does End-To-End Training Fail?](https://arxiv.org/abs/1902.04422)

**Questions:**

- Do you think it would be beneficial to drop a different fraction of base learners each step during the joint training of the ensemble to avoid learner codependency?
- Minor comments:
  - As mentioned above, the formatting of the reference is incorrect. At least the title of the paper should be visible.
  - NeurIPS style rules do not allow vertical lines in tables. I would recommend removing the background color as well.
  - Overall, I like the paper, but you spend too much space organizing and formalizing previous research; I think it would be better to add the experiments in the appendix to the main text.
  - In Figure 6, it looks better to add the ImageNet title to the two graphs on the right for consistency.

**Limitations:**

I see no potential negative societal impact from this paper.

---

> ### Author Rebuttal · Authors · 2023-08-09
>
> We thank this reviewer for their constructive feedback. We have addressed the points raised below which have helped us to further improve our submissions clarity and contribution.
>
>   $~$
>
> * _Practical contribution_ - While we appreciate that the focus of this work was not on proposing new methods, we do believe that identifying and characterizing the limitations of joint training will still have a significant practical impact. We have already discussed several previous works which have directly attempted joint training (see Sec. 2 “joint training” & a deeper exposition of the limitations of the analyses of these works in App E) but it is likely that this issue is regularly unknowingly rediscovered by practitioners as joint training is a natural objective to consider. For example: (a) [1] reports training an ensemble sequentially due to “unstable” training dynamics when training simultaneously, (b) [2] introduces a regularization term to prevent their ensemble from “collapsing to degenerate solutions” and (c) [3] which argues that training for both loss and diversity on the same data may "render the convergence point of the training process uncontrollable". We hope that, at a minimum, our work can act as a canonical reference for this issue such that future works might avoid this common pitfall.
>
>   $~$
>
>   Regarding the augmented objective, we would point out that our objective is more general than that of Webb et. al as it is defined for all twice or more differentiable loss functions. However, we agree that as a proposed solution this would be similar in spirit. In our work, we intended the augmented objective to primarily be a tool for analysis of the limitations of joint training rather than a complete solution. We do hope that the result indicating that low levels of $\beta$ still performing reasonably well will inspire future methods that can overcome the limitations we have described.
>
>   $~$
>
>   **Action Taken**: We have added a note on the consequences of learner collusion to future ensemble research in our conclusion.
>
>   $~$
>
>   [1] Pagliardini, Matteo, et al. "Agree to Disagree: Diversity through Disagreement for Better Transferability." The Eleventh International Conference on Learning Representations. 2022.
>
>   [2] Lee, Yoonho, Huaxiu Yao, and Chelsea Finn. "Diversify and disambiguate: Out-of-distribution robustness via disagreement." The Eleventh International Conference on Learning Representations. 2022.
>
>   [3] Pang, Tianyu, et al. "Improving adversarial robustness via promoting ensemble diversity." International Conference on Machine Learning. PMLR, 2019.
>
>   $~$
>
> * _dropping a fraction of base learners each step during the joint training_ - We agree that this is a very natural solution to learner collusion, thank you for the suggestion! The hypothesis being that if we drop a sufficiently large portion of base learners in, say, each batch but still perform joint training on the remainder, this should cause inflated diversity to be actively harmful even on the training data and therefore force the ensemble to avoid this degeneracy. We decided to try out this idea which we describe next.
>
>   $~$
>
>   We repeat the setup on CIFAR-10 with ResNet-18 architecture as described in the paper. We train each model using the joint objective, but at each batch we drop a proportion $p \in [0, 0.2, 0.4, 0.6]$ of randomly selected learners from the ensemble. We then investigate if this (a) reduces collusion and (b) improves ensemble performance. **The results of this experiment are included in the attached pdf**. We find that dropping learners does indeed significantly reduce the diversity score indicating a reduction in learner collusion (although a reasonably large proportion of learners is required to be dropped). Unfortunately, the improvement in performance by reducing collusion is negated by a decrease in individual performance due to the base learners becoming weaker on average. The reason why is apparent once we notice that by dropping a base learner from some proportion of batches, this is exactly equivalent to bootstrapping. While bootstrapping has been effective for ensembles of weak learners (e.g. random forest) it has already been shown to be harmful for deep ensembles [4]. Unfortunately, despite being a sensible idea, this resolution can only reduce the effect of learner collusion by simultaneously harming the performance of the ensemble due to bootstrapping.
>
>   $~$
>
>   Proposing methods to overcome learner collusion is a valuable but non-trivial direction for future work that we and, we hope, others in the research community will pursue. We believe that the comprehensive diagnosis of the issue we have presented in this work will provide a foundation for future methodological progress.
>
>   $~$
>
>   **Action taken**: We have added the results of this experiment as a negative result in order to aid future research.
>
>   $~$
>
>   [4] Nixon, Jeremy, Balaji Lakshminarayanan, and Dustin Tran. "Why are bootstrapped deep ensembles not better?." ''I Can't Believe It's Not Better!''NeurIPS 2020 workshop. 2020.
>
>   $~$
>
> * _Formatting issues_ - We wish to sincerely apologize to all four reviewers for a (silent) latex compiling error in the final version of our manuscript that resulted in the titles of the papers not appearing in the PDF. This was an unfortunate inconvenience for the reviewers. We do wish to highlight that several reviewers made positive comments about the writing and organization of this work – thus, the issues regarding the presentation are limited to simple **formatting errors which we have now rectified**.  Furthermore, we have implemented the further formatting suggestions from this reviewer and, should the paper be accepted, would ensure careful attention is paid during further polishing of the manuscript for a camera-ready version.

---

> > ### Comment · Reviewer_pZUZ · 2023-08-12
> >
> > Thank you for the answers and additional experiments. I find this finding very interesting and think it will be helpful for future research. I have decided to raise my review score.

---

> > > ### Author Response · Authors · 2023-08-15
> > >
> > > We are delighted we were able to address this reviewer's concerns and appreciate their positive conclusion. We thank them for a constructive review process which helped improve our paper's clarity and contribution.

---

### Official Review · Reviewer_m1Sy · 2023-07-06

**Soundness:** 3 good
**Presentation:** 4 excellent
**Contribution:** 2 fair
**Rating:** 4
**Confidence:** 4

**Summary:**

This paper explores the joint training of deep ensembles, wherein the ensemble error is directly optimized during training. The authors find that joint training leads to poor performance, which they posit it due to phenomenon they call "learner collusion", where base learners artificially inflate their diversity, at the expense of test performance. The authors provide a new decomposition of a broad class of loss functions in terms of the average error rate, the ensemble error rate, and a generalized notion of diversity. The authors then perform an experimental evaluation into the prevalence of learner collusion during joint training, using an augmented objective function to probe the effect of explicitly encouraging diversity during training.

**Strengths:**

Overall, the paper is very well-written and easy to follow, and in my view the topic is very relevant to the community.

To me, the most compelling experimental evidence for the learner collusion phenomenon are those presented under "learner codependence"/in Figure 4. From these results it is very clear that something distinct is happening in the joint training regime, and that the individual models are exploitating the joint loss. The definition of the augmented objective $\mathcal{L}^\beta$ seems like a very useful tool for probing these and other related phenomena.

**Weaknesses:**

Unless I am misunderstanding, the conclusions drawn from Figure 2 do not seem to strongly support the claim that joint training significantly harms ensemble performance. Indeed, for the Facebook task, it seems the method actually helps, while for others it does not. This is in contrast to the CIFAR results in Table 1/Figure 6, where the degredation in performance is significant. Could the authors provide some clarification as the intended conclusion from these experiments?

There are a number of other papers in the literature that study some form of the gap (in the authors' notation) $\bar{\text{ERR}} - \text{ERR}$, and express and/or bound it in terms of various forms of "diversity", depending on the given loss function. For example Ortega et al., 2022, Abe et al., 2022 and Masegosa et al., 2020 give explicit expressions for the average - ensemble error gap in terms of diversity metrics that are easily interpretable. It's unclear to me that the form in Theorem 4.5 adds much to this literature, and indeed the stated expression for diversity doesn't seem to be used in the remainder of the paper.

Overall, while I think the paper is well-written and the topic is relevant, I think there is 1) insufficient use/novelty in the theoretical results and 2) somewhat incomplete conclusions to be drawn from the empirical results to recommend accepting at the the current stage. In particular, while it certainly appears like some type of "learner collusion" is happening, it seems very important to understand why this only appears at $\beta=1$, and exactly what impact this could have on the ensemble error.

As a minor point, there seems to be an issue with the references -- they don't seem to contain any titles of the referred papers.

**References:**

Taiga Abe, E Kelly Buchanan, Geoff Pleiss, and John Patrick Cunningham. The best deep
ensembles sacrifice predictive diversity, 2022.

Luis A. Ortega, Rafael Cabañas, and Andres Masegosa. Diversity and generalization in neural
network ensembles, 2022.

Andres Masegosa, Stephan Lorenzen, Christian Igel, and Yevgeny Seldin. Second order PAC-
Bayesian bounds for the weighted majority vote, 2020.

**Questions:**

I find the distinct change in behavior of the various metrics at $\beta=1$ very interesting; it seems the behavior is relatively benign for $\beta < 1$, and yet there appears to be a distinct transition at $\beta = 1$. Do the authors have any hypotheses as to why this occurs?

---

> ### Author Rebuttal · Authors · 2023-08-06
>
> We thank this reviewer for their constructive feedback. We have addressed the points raised below which have helped us to further improve our submissions clarity and contribution.
>
> $~$
>
> * _Facebook dataset & Formatting_ - Please see our response in the general comment section.
>
> $~$
>
> * _On Thm. 4.5_ - We apologize for not sufficiently emphasizing the significance of this result. This theorem is an essential ingredient as it demonstrates that, for any at least twice differentiable loss function, the ensemble loss can be decomposed into the aggregate individual loss and a term that captures ensemble diversity. This generalizes specific examples of this decomposition (e.g. Krogh and Vedelsby, 1994) and unifies any analysis of joint training to practically any loss function. Indeed, we do not manually calculate diversity using the Hessian as it is more efficiently calculated as $\bar{ERR} - ERR$, but having this theorem ensures that the term we obtain from this calculation can always be interpreted as ensemble diversity. Furthermore, this theorem also guarantees that for positive semi-definite losses (e.g. MSE, Cross-Entropy), the diversity is lower bounded by zero – an important property for a sensible definition of diversity.  We emphasize that without this theorem we could only make conclusions about specific loss functions whilst with it we can discuss diversity and joint training in the general setting – resulting in us being the first paper to do so.
>
>   $~$
>
>   **Action taken**: We have added further context on the significance of this result in the text.
>
>   $~$
>
>
> * _Distinct transition at $\beta = 1$_ - This is indeed a notable phenomenon that we only addressed in the appendix. Let us provide a more succinct discussion on this point.
>
>   $~$
>
>   Figures 2-4 do highlight a sudden jump in learner collusion as $\beta \to 1$ in the case of regression rather than a gradual increase.  Why does this occur? A useful explanation lies in the gradient-adjusted target (GAT) analysis in App D. To briefly summarize, this section considers applying individual training but with an adjusted target which is a function of the gradient of the ensemble error. Specifically, we adjust the targets to account for the errors of the ensemble. Formally, we optimize $\frac{1}{M}\sum_{j=1}^M (f_j - \bar{y})^2$ where $\bar{y}$ is the adjusted target given by $\bar{y} = y - \alpha \cdot g$. Here $g$ is the ensemble loss gradient and $\alpha$ is a step size parameter determining how big a step the target of an individual learner should take in order to account for the errors of the ensemble. In Thm. D.1 we show that this GAT objective is exactly equivalent to our augmented objective in Sec. 5 with $\beta = (1 - \frac{1}{(1 + \alpha)^2})$. Given this relationship, we might notice that the gradient step $\alpha$ grows exponentially as $\beta \to 1$ resulting in each individual learner having their targets biased excessively to account for the ensemble errors (**please see the new figure in the attached PDF visualizing this relationship**). Given this perspective, it is unsurprising that learner collusion does not grow linearly with $\beta$.
>
>   $~$
>
>   **Action taken**: We have added a succinct summary of this point in the main text. We have also added this new plot and refined the discussion in App D.
>
> $~$
>
> * _Related works_ - Thank you for raising this point, we hope that extending our discussion to these works will further highlight our contribution. Firstly, we wish to reiterate that our work investigates the direct optimization of the loss of the ensemble which we refer to as joint training. Given our findings (i.e. learner collusion), it is not surprising that there have been several works that attempt to improve ensemble performance by other means (i.e. proposing alternative methods of encouraging diversity). We categorized this literature as “ensemble-aware individual training” in our background section. We believe that our investigation of the underlying issue with directly optimizing the ensemble will provide an important basis for future methods of this type.
>
>   $~$
>
>   Masegosa et al. (2020) is one such example in which the authors propose to optimize a PAC-Bayesian generalization bound in the case of cross-entropy. Whilst conceptually appealing, optimizing the bounds of model performance is generally not a standard approach in machine learning as it is unclear how minimizing a worst-case bound or maximizing a best-case bound directly affects the actual generalization performance under the metric of interest of a model. However, given the limitations we have identified in directly optimizing that metric of interest in our work, this approach might offer an effective alternative providing a fruitful direction for future work. Then Ortega et al. (2022) extended this approach to additional loss functions, analytical analysis and a more practical empirical evaluation. We hope that our contribution here will provide a useful foundation for alternative ensemble training approaches such as these.
>
>   $~$
>
>   The recent workshop paper of Abe et al. (2022) is indeed relevant as it considers decomposing the objective into individual loss and diversity for two specific cases (MSE and CE with probability averaging) and provides a useful motivation for our investigation of joint training. Whilst this work does provide preliminary evidence of poor performance under joint training (an empirical observation that was also reported and misdiagnosed previously in the literature as discussed extensively in our related work and App. E), it does not investigate why this occurs - the primary contribution of our work. Furthermore, all of the contributions of our work (as summarized in Fig 7), are entirely novel with respect to this work.
>
>   $~$
>
>   **Action taken**: We have integrated these works into our related work discussion, thank you for encouraging us to extend our literature review.

---

> > ### Comment · Senior_Area_Chairs · 2023-08-18
> > **@m1Sy: Please engage with rebuttal**
> >
> > The authors have posted their rebuttal to your review—does it affect your opinion? It'd be helpful if you can at least acknowledge having read the rebuttal, even if you don't find it convincing. Thanks!

---

### Official Review · Reviewer_XMTw · 2023-07-07

**Soundness:** 2 fair
**Presentation:** 1 poor
**Contribution:** 3 good
**Rating:** 5
**Confidence:** 4

**Summary:**

This paper mainly studies the reason behind the failure of jointly training deep ensembles. It discovers that joint optimization results in a phenomenon in which base learners collude to artificially inflate their apparent diversity. Both theoretical and empirical evidence are provided further to verify the hypothesis.

**Strengths:**

(1) This paper attempts to study a seemingly under-explored question, that is, why joint training of ensembles fails to generalize better than individual training and ensemble. This research topic is very important and helps us to understand the foundations of the deep ensemble.

(2) Both theoretical proof and empirical proof are provided.

**Weaknesses:**

(1) One big issue of this paper is that all references do not contain titles, which to be honest, I have never seen in top-tier conferences like NeurIPS.

(2) I do not understand why joint training is easier to train but results in poor generalization. What does "easier to train" refer to?

(3) What joint training approaches are used in the paper?

(4) The empirical evaluation isn't very convincing to me. For instance, how many ensemble members are we using for "Learner codependence"? It is not surprising that the performance degrades to me if we drop ensemble members during testing, especially if our overall ensemble members is not sufficient.

**Questions:**

Please see my above weaknesses.

---

> ### Author Rebuttal · Authors · 2023-08-09
>
> We thank this reviewer for their constructive feedback. We have addressed the points raised below which have helped us to further improve our submissions clarity and contribution.
>
>   $~$
>
> * _What joint training approaches are used_ - Joint training refers to the case when the aggregated predictions of the entire ensemble are optimized directly. Formally (using the notation from our paper): given an ensemble of learners $f_1, \ldots, f_M$, we might consider optimizing the _joint objective_ $\mathcal{L}(\frac{1}{M} \sum_{i=1}^{M} f_i , y)$ ($= \text{ERR}$) where $\mathcal{L}$ is a given loss function such as mean squared error or negative log-likelihood. This is in contrast to the commonly observed _independent training_ where each ensemble member is optimized directly resulting in an optimization objective given by $\frac{1}{M} \sum_{i=1}^{M} \mathcal{L}(f_i , y)$ ($= \bar{\text{ERR}}$).
>
>   $~$
>
>   **Action taken**: This definition is an important foundation for the contributions of our paper. We have therefore added this formal definition earlier to the introduction section to ensure it is established up front and without ambiguity.
>
>   $~$
>
> * _What does "easier to train" refer to?_ - We did not make any claims regarding any method being “easier to train”. In fact, this quote does not appear in our paper. A point that we did make (e.g. L30-38) is that – despite the joint objective being the “true objective of interest” – throughout the literature, we typically observe the ensemble being trained independently in practice (occasionally with some regularization on the ensemble). Again, let us make this point more formally using the same notation as the previous point. The key question our paper addresses is: given that when we evaluate the performance of an ensemble (e.g. at test time or in production) using the loss term $\text{ERR}$, why do we not optimize for that objective during training rather than the commonly observed proxy of $\bar{\text{ERR}}$? Of course, the conclusion is that this is for good reason: the joint training of deep ensembles (i.e. optimizing $\text{ERR}$) fails due to learner collusion.
>
>   $~$
>
>   **Action taken**: We have integrated this point more formally into the introduction as part of the additional formalism from our previous action point. Thank you for encouraging us to further improve our clarity.
>
>   $~$
>
> * _Empirical evaluation_ - We agree that extending the empirical evaluation of the practical limitations of joint training can further strengthen this work. We have therefore **extensively broadened our evaluation on ImageNet** (from Table 1 and Figure 6 RHS) with eight additional architectures and various ensemble sizes. All models are trained from scratch and **we include these new results in the attached PDF document** (Table 5) where we consistently find that joint training performs significantly worse than independent training, thus reinforcing the claims of our work.
>
>   $~$
>
>   **Action taken**: We appreciate this suggestion and have updated our manuscript by including these additional results in Sec. 3.
>
>
>   $~$
>
> * _It is not surprising that the performance degrades to me if we drop ensemble members during testing (regarding Figure 4)_ - Indeed, it is correct to assume that dropping ensemble members at test time will likely result in a degradation in performance. Our intention in this experiment was to analyze the extent of that degradation for different training methods. If joint training results in learner collusion as we hypothesize, we would expect that the individual learners would become more codependent in that setting. To illustrate this, consider the trivial case of two identical learners that reasonably accurately predict the label $f_1 = f_2 \approx y$. Should they collude by biasing their regression predictions in opposing directions by some constant $k \in \mathbb{R}$ (i.e. $f_1^\text{bias} = f_1 + k$ and $f_2^\text{bias} = f_2 - k$), this would artificially inflate diversity ($\text{DIV} = \frac{1}{2} \sum_{j=1}^2(\bar{f} - f_j^\text{bias})^2 = k^2$) but result in a codependence such that dropping either learner at test time would be catastrophic to the ensemble performance as the remaining learner would be highly biased by exactly the term $k$. If these learners were independently trained and, therefore, not colluding, this bias can not be learned and the resulting drop in performance should be significantly less. In Figure 4 this is exactly what we observe. For all values of $\beta$ we consider the relative increase in test loss upon dropping a subset of the learners at test time. As $\beta \to 1$ (i.e. as we approach joint training) we observe a very large increase in that test error while elsewhere the relative increase in error is far more modest.
>
>   $~$
>
>   **Action taken**: We have added a short interpretation of the results of this experiment directly in the caption of Figure 4. Again, thank you for encouraging us to further clarify exposition.
>
>   $~$
>
> * _Formatting issue_ -  We wish to sincerely apologize to all four reviewers for a (silent) latex compiling error in the final version of our manuscript that resulted in the titles of the papers not appearing in the PDF. This was an unfortunate inconvenience for the reviewers. We do wish to highlight that several reviewers made positive comments about the writing and organization of this work – thus, the issues regarding presentation are limited to simple **formatting errors which we have now rectified**. Should the paper be accepted, we would ensure careful attention is paid during further polishing of the manuscript for a camera-ready version.

---

> > ### Comment · Senior_Area_Chairs · 2023-08-18
> > **@XMTw: Please engage with rebuttal**
> >
> > The authors have posted their rebuttal to your review—does it affect your opinion? It'd be helpful if you can at least acknowledge having read the rebuttal, even if you don't find it convincing. Thanks!

---

> > ### Comment · Reviewer_XMTw · 2023-08-20
> > **Thanks for the response**
> >
> > I thank the authors for the detailed response. It addressed some of my concerns. I believe the empirical results can contribute to the community. However, I believe the primary finding of this paper, i.e., jointly trained ensemble leads to learner Collusion is somehow trivial and straightforward. It is natural for me that the jointly trained ensemble leads to collusion without adding any regularization. Therefore, I would like to increase my score to 5, not 6.

---

> > > ### Author Response · Authors · 2023-08-20
> > >
> > > We thank this reviewer for their response and are glad that they have determined that the paper should be accepted. We agree that their suggested extension to our empirical results reinforces the claims of our work.
> > >
> > > Let us briefly address the point that our findings are "*somehow trivial and straightforward*" as this was not raised in the original review (indeed, there this reviewer stated that "*this research topic is very important and helps us to understand the foundations of the deep ensemble*"). Our goal was to present this phenomenon as clearly and intuitively as possible and we are glad it was perceived as such. However, despite being a natural explanation in retrospect, the limitations of jointly training ensembles has been a **reoccurring issue** throughout the literature and, in the rare cases where they have been investigated, they have been **misdiagnosed**. We therefore suggest that a clear investigation of the issue, as provided by this work, is a worthwhile contribution to the literature which can (a) prevent the need to constantly rediscover this degeneracy, (b) amend the previous literature, and (c) guide future research into the optimization of deep ensembles.
> > >
> > >  $~$
> > >
> > > * **Reoccurring issue** - It is likely that this issue is regularly and unknowingly rediscovered by practitioners as joint training is a natural objective to consider. For example: (a) [1] reports training an ensemble sequentially due to “unstable” training dynamics when training simultaneously, (b) [2] introduces a regularization term to prevent their ensemble from “collapsing to degenerate solutions", (c) [3] states that training for both loss and diversity on the same data may "render the convergence point of the training process uncontrollable", and (d) [4] mentions that joint training "reduces the accuracy of the ensemble and can easily lead to training instabilities". We hope that, at a minimum, our work can act as a canonical reference for this issue such that future works might avoid this common pitfall.
> > >
> > > * **Misdiagnosed** - A small number of works have previously hypothesized about this problem. We addressed these works in Sec. 2 “joint training” with a deeper analysis in App E. The key takeaway is that the existing attempts to address this issue are insufficient for explaining the general phenomenon or, in some cases, are objectively incorrect (e.g. [5] reported successful joint training under SoftMax averaging which was discovered to be a coding bug that resulted in unintentionally implementing independent training instead). Although this degeneracy might appear trivial to this reviewer, we believe that there has been sufficient confusion in the literature to warrant our comprehensive analysis.
> > >
> > >  $~$
> > >
> > > [1] Pagliardini, Matteo, et al. "Agree to Disagree: Diversity through Disagreement for Better Transferability." The Eleventh International Conference on Learning Representations. 2022.
> > >
> > > [2] Lee, Yoonho, Huaxiu Yao, and Chelsea Finn. "Diversify and disambiguate: Out-of-distribution robustness via disagreement." The Eleventh International Conference on Learning Representations. 2022.
> > >
> > > [3] Pang, Tianyu, et al. "Improving adversarial robustness via promoting ensemble diversity." International Conference on Machine Learning. PMLR, 2019.
> > >
> > > [4] Mehrtens, Hendrik Alexander, Camila Gonzalez, and Anirban Mukhopadhyay. "Improving robustness and calibration in ensembles with diversity regularization." DAGM German Conference on Pattern Recognition. Cham: Springer International Publishing, 2022.
> > >
> > > [5] Dutt, Anuvabh, Denis Pellerin, and Georges Quénot. "Coupled ensembles of neural networks." Neurocomputing 396 (2020): 346-357.

---

> > > > ### Comment · Reviewer_XMTw · 2023-08-20
> > > > **Follow-up**
> > > >
> > > > I thank the authors for the quick response. I really appreciate their hard work. I believe that increasing the score to 5 does not necessarily means that I "have determined that the paper should be accepted". Moreover, I don't think it is not common to add new comments that were not raised in the original review. Aftering reading the new response, I am still not convinced that the findings in this paper are novel and non-trivial enough.

---

> > > > > ### Author Response · Authors · 2023-08-20
> > > > >
> > > > > Likewise, we sincerely appreciate this reviewer's engagement. Regarding the new point raised, we only meant to clarify that since this was a new comment it was not something we had already addressed in our original rebuttal and would therefore respond in that comment.
> > > > >
> > > > > We suggest that despite the intuitive simplicity of base learners collectively biasing their prediction to artificially inflate diversity (i.e. learner collusion), this work may be of significant interest to others in the community where this effect has not been previously identified and, in fact, the question of *why* and *when* joint training fails has been misdiagnosed. For *why* it occurs, we note that the previous work of [6] who also noticed issues in joint training claimed that it was due to the fact that "gradients back-propagated into all ensemble members are identical". At the time of writing, we note that this work has been cited 270 times -- indicating that this issue is relevant to a reasonable subset of researchers. Then, in Appendix E, we showed that this explanation cannot explain the issue as it is not true (as the authors claim) in the case of probability averaging. Furthermore for *when* it occurs, we unify the joint objective decomposition to all twice differentiable loss functions rather than just cross-entropy loss which connects the issues in the regression setting with those of the classification setting for the first time.
> > > > >
> > > > > This is in addition to our other contributions which include (1) a tighter upper bound on diversity for guaranteed divergence for diversity, (2) a general augmented diversity objective with connections made to individual training with a gradient-adjusted target (i.e. GAT objective in App D), and (3) extensive novel empirical analysis (i.e. comparisons of joint training and independent training, analysis of increasing diversity, analysis of training dynamics, analysis of ensemble size, score-averaging vs probability averaging, and consideration of $\overline{\text{ERR}}$ as a metric instead of accuracy). We sincerely believe that providing the first accurate characterization of why joint training leads to this degeneracy in addition to the other aforementioned contributions will be valuable to a subset of the community who are developing novel theory and methods for ensembling deep neural networks.
> > > > >
> > > > > Your comment has helped us appreciate that these contributions could be made clearer in our text for future readers and, should this work be accepted, we would endeavor to improve our description of the contributions of our work early in the text.

---

### Author Rebuttal · Authors · 2023-08-09

We thank all four reviewers for their constructive feedback. We have found their feedback to be instructive with their suggestions and questions helping us to further improve our submissions clarity and contribution.

  $~$

* _Formatting error_ - We wish to sincerely apologize to all four reviewers for a (silent) latex compiling error in the final version of our manuscript that resulted in the titles of the papers not appearing in the PDF. This was an unfortunate inconvenience for the reviewers. We do wish to highlight that several reviewers made positive comments about the writing and organization of this work – thus, the issues regarding the presentation are limited to **simple formatting errors which we have now rectified**.  Furthermore, we have implemented further formatting suggestions from pZUZ and, should the paper be accepted, would ensure careful attention is paid during further polishing of the manuscript for a camera-ready version.

  $~$

* _Clarification on Figure 2 (Facebook dataset)_ - We thank reviewers **m1Sy** & **a4xj** who highlighted that it would be beneficial to comment on this result, we certainly agree that this particular result requires some further clarification. Unlike all other experiments throughout this paper, the test error on this task is relatively flat across all values of $\beta$ (after accounting for the standard errors). Therefore, one might ask whether this result contradicts any of our claims.

  $~$

  We begin by revisiting our analytical analysis of the regression case in App F. While this section mathematically uncovers how optimizing for diversity results in diversity-inflating bias terms, it does not guarantee that it will result in worse test performance. It is at least conceptually possible to construct a task in which diversity is not desirable and, therefore, learner collusion is not harmful. Further analysis indicates that this dataset is such a case (we also highlight that this is quite a popular regression task in the machine learning literature e.g. [1-3]). The goal in this task is to predict the number of impressions a post will obtain given several metrics. While this is a challenging task due to high variance in the response, the signal that does exist seems to be almost entirely contained within a single variable: “number of likes” resulting in no need for true diversity and, therefore, no substantial impact due to learner collusion. We verify this claim by matching the reported test performance with a single decision tree of depth 3 which (a) matches the performance of a neural network (MSE = 2.18 $\pm$ 0.66 over 5 runs) and (b) has feature importance dominated by this single “number of likes” variable (in all 5 runs the tree uses this feature as its primary split).

  $~$

  Finally, we note that this dataset is still valuable as (1) this experiment was investigating the existence of learner collusion – which certainly still occurs on this task, and (2) selective inclusion of datasets based on nice characteristics should be avoided – indeed, we hope that this result and our accompanying clarification may be instructive for future readers. Readers might also be interested to note that we have **extensively broadened our evaluation on ImageNet** (from Table 1 and Figure 6 RHS) with eight additional architectures and various ensemble sizes. All models are trained from scratch and **we include these new results in the attached PDF document** (Table 5) where we consistently find that joint training performs significantly worse than independent training, thus reinforcing the claims of our work.

  $~$

  **Action taken**: We have added a comment to the paper clarifying this result and significantly extended our experiments on ImageNet.

  $~$

  [1] Romano, Yaniv, Evan Patterson, and Emmanuel Candes. "Conformalized quantile regression." Advances in neural information processing systems 32 (2019).

  [2] Sesia, Matteo, and Yaniv Romano. "Conformal prediction using conditional histograms." Advances in Neural Information Processing Systems 34 (2021): 6304-6315.

  [3] Jeffares, Alan, et al. "TANGOS: Regularizing Tabular Neural Networks through Gradient Orthogonalization and Specialization." The Eleventh International Conference on Learning Representations. 2022.


  $~$

---

### Decision · Program_Chairs · 2023-09-21

**Decision:**

Accept (poster)

**Comment:**

While there was something of a split in scores on this paper, the balance of sentiment seems to me clearly in favor of acceptance, especially given that the lowest score was given by a reviewer who did not engage with the rebuttal (which, in my opinion and that of at least some of the other reviewers during later discussions, addresses many of that reviewer's concerns). While on one level I agree with Reviewer XMTw that it's not shocking that jointly trained deep ensembles learn to collude in a problematic way, demonstrating and explaining that phenomenon cleanly and clearly is a significant and valuable contribution.